# Identification of a Munc13-sensitive step in chromaffin cell large dense-core vesicle exocytosis

Kwun Nok M Man[1], Cordelia Imig[1], Alexander M Walter[2], Paulo S Pinheiro[3], David R Stevens[4], Jens Rettig[4], Jakob B Sørensen[3], Benjamin H Cooper[1], Nils Brose[1], Sonja M Wojcik[1]*

[1]Department of Molecular Neurobiology, Max Planck Institute of Experimental Medicine, Göttingen, Germany; [2]Leibniz Institute for Molecular Pharmacology, Berlin, Germany; [3]Department of Neuroscience and Pharmacology, Faculty of Health and Medical Sciences and Lundbeck Foundation Center for Biomembranes in Nanomedicine, University of Copenhagen, Copenhagen, Denmark; [4]Department of Physiology, Saarland University, Homburg, Germany

**Abstract** It is currently unknown whether the molecular steps of large dense-core vesicle (LDCV) docking and priming are identical to the corresponding reactions in synaptic vesicle (SV) exocytosis. Munc13s are essential for SV docking and priming, and we systematically analyzed their role in LDCV exocytosis using chromaffin cells lacking individual isoforms. We show that particularly Munc13-2 plays a fundamental role in LDCV exocytosis, but in contrast to synapses lacking Munc13s, the corresponding chromaffin cells do not exhibit a vesicle docking defect. We further demonstrate that ubMunc13-2 and Munc13-1 confer $Ca^{2+}$-dependent LDCV priming with similar affinities, but distinct kinetics. Using a mathematical model, we identify an early LDCV priming step that is strongly dependent upon Munc13s. Our data demonstrate that the molecular steps of SV and LDCV priming are very similar while SV and LDCV docking mechanisms are distinct.

*For correspondence: wojcik@ em.mpg.de

Competing interests: The authors declare that no competing interests exist.

## Introduction

The regulated, $Ca^{2+}$-triggered secretion of catecholamines from chromaffin cell LDCVs is an integral part of the physiological adaption to environmental stressors. Like the exocytosis of neuronal SVs, LDCV exocytosis is mediated by SNARE complex formation, in concert with $Ca^{2+}$ sensors and essential regulatory proteins (*James and Martin, 2013*; *Neher, 2006*; *Ovsepian and Dolly, 2011*; *Pang and Sudhof, 2010*).

Mammalian uncoordinated 13 (Munc13) proteins are essential SV priming factors in neurons (*Augustin et al., 1999*; *Richmond et al., 1999*; *Rosenmund et al., 2002*), and ultrastructural studies have shown that in synapses lacking Munc13s/Unc-13, SVs also fail to physically dock to synaptic active zones (*Imig et al., 2014*; *Siksou et al., 2009*; *Weimer et al., 2006*). At the molecular level, this morphological phenotype most likely corresponds to a role of Munc13s in mediating the formation of SNARE complexes at vesicular release sites (*Hammarlund et al., 2007*; *Hammarlund et al., 2008*; *Imig et al., 2014*; *Ma et al., 2011*; *2013*; *Yang et al., 2015*).

The Munc13 family consists of five members, Munc13-1 (*Unc13a*), Munc13-2 (*Unc13b*), Munc13-3 (*Unc13c*), the brain specific angiogenesis inhibitor I-associated protein 3 (*Baiap3*), and the non-neuronal isoform Munc13-4 (*Unc13d*) (*Koch et al., 2000*). Genetic deletion of *Unc13a* and *Unc13b* completely eliminates SV exocytosis in hippocampal neurons (*Varoqueaux et al., 2002*), and selectively reduces synaptic vs. extrasynaptic exocytosis of neuronal LDCVs (*van de Bospoort et al.,*

**eLife digest** Mammals have adrenal glands, which secrete the stress hormone adrenaline as well as other hormones into the bloodstream. These molecules are produced in chromaffin cells, where they are packaged into compartments called large dense-core vesicles (LDCVs). To release the hormones into the bloodstream, the vesicles bind to and fuse with the membrane that surrounds the cell. This process – which is called exocytosis – is triggered by increases in the level of calcium ions inside the cells.

Exocytosis also enables nerve cells to release chemical signals at junctions (known as synapses) with other nerve cells. These signals are packaged within another type of vesicle called 'synaptic' vesicles, which also release their contents by fusing with the cell membrane. However, it is not clear whether the two types of vesicle carry out exocytosis in the same way.

Exocytosis requires that the vesicles physically attach to the membrane and undergo a process termed 'priming', which enables them to fuse quickly with the membrane in response to an increase in calcium ion levels. In synaptic vesicles, both of these processes – physical membrane attachment and priming – appear to occur in a single step that requires a family of proteins called the Munc13 proteins. Here, Man et al. investigate whether the Munc13 proteins are also essential for LDCV exocytosis in the chromaffin cells of mice. The experiments reveal that in contrast to synaptic vesicles, the initial binding of LDCVs to membranes does not require Munc13 proteins. However, the loss of one member of the family called Munc13-2 dramatically reduces the fusion of LDCVs with the membrane of chromaffin cells. Further experiments reveal that different Munc13 proteins differ in their ability to drive the exocytosis of LDCVs.

Man et al. use a mathematical model of LDCV exocytosis, which reveals that Munc13 plays an important role in the first part of the priming step. Together, these findings show that synaptic vesicles and LDCVs use different mechanisms to bind to membranes, but are primed for fusion in a similar way.

*2012*), which indicates that SV and LDCV exocytosis at active zones is mediated by similar molecular mechanisms. By contrast, studies in *C. elegans* and *Drosophila* have shown that Unc-13/dUnc-13 selectively regulate SV release, whereas the $Ca^{2+}$-dependent activator proteins for secretion (CAPS/Unc-31) specifically regulate LDCV release (*Hammarlund et al., 2008*; *Renden et al., 2001*; *Speese et al., 2007*; *Zhou et al., 2007*).

In mammals, Munc13s and CAPSs appear to perform non-redundant functions critical for both SV and LDCV exocytosis in neurons (*Jockusch et al., 2007*; *van de Bospoort et al., 2012*), as well as for LDCV exocytosis in neuroendocrine cells (*Elhamdani et al., 1999*; *Kabachinski et al., 2014*; *Kang et al., 2006*; *Kwan et al., 2006*; *Liu et al., 2010*; *Liu et al., 2008*; *Speidel et al., 2008*). Yet, to date, while CAPS-1 and CAPS-2 have been shown to be required for LDCV exocytosis in mammalian chromaffin cells (*Liu et al., 2010*; *Liu et al., 2008*), evidence that endogenous Munc13s are required for LDCV exocytosis is lacking. In fact, the role of Munc13-1 and ubMunc13-2 has only been examined in the context of overexpression studies, and other isoforms have not been investigated (*Ashery et al., 2000*; *Bauer et al., 2007*; *Liu et al., 2010*; *Stevens et al., 2005*; *Zikich et al., 2008*).

In the present study, we performed the first comprehensive analysis of all neuronal and neuroendocrine members of the Munc13 protein family in chromaffin cells, defining their respective roles in LDCV exocytosis. We identify the $Ca^{2+}$-dependent step in the priming process at which Munc13-1 and ubMunc13-2 operate, and demonstrate that, although they are critical for LDCV priming and release, LDCV docking can occur without them.

## Results

### Expression of Munc13 isoforms in the mouse adrenal gland

We first analyzed the expression of all Munc13 isoforms in the murine adrenal gland by western blotting (*Figure 1*). In perinatal adrenal glands, we detected Munc13-1 (*Figure 1A* and *Figure 1—figure*

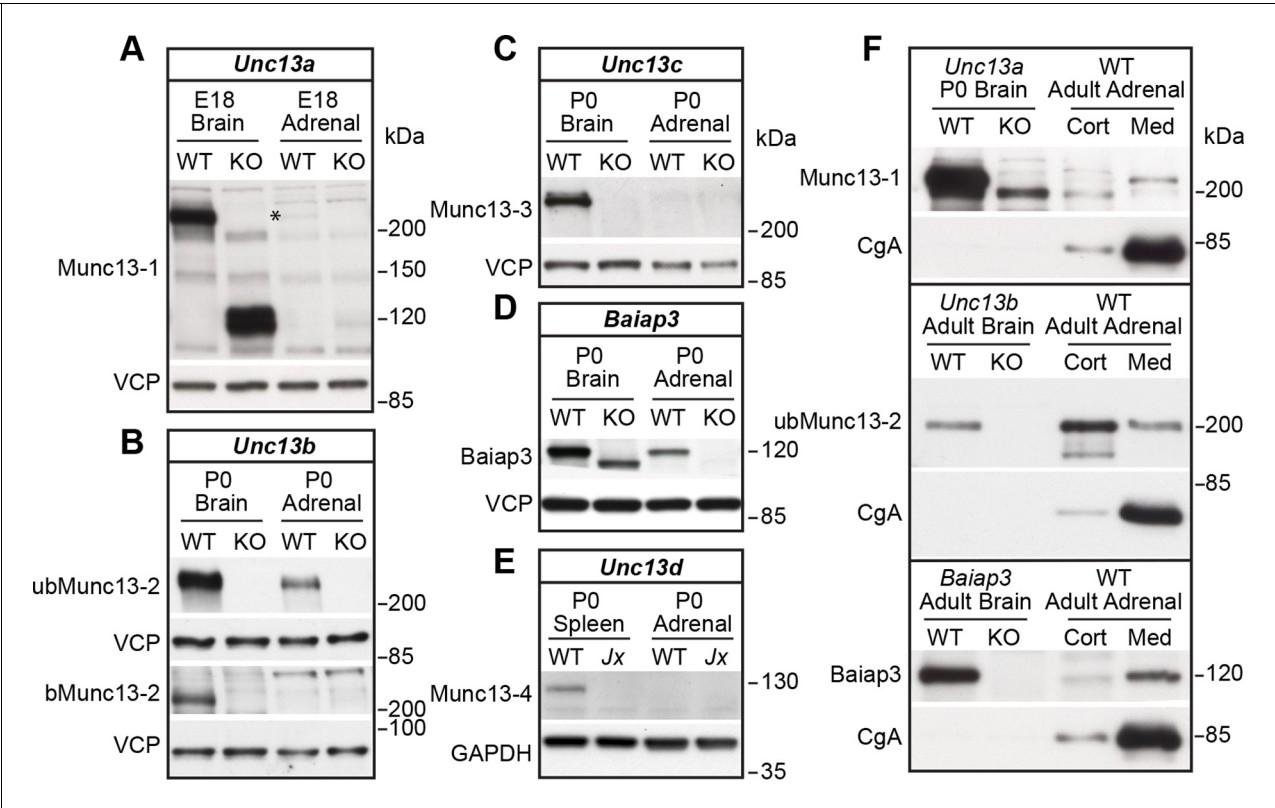

**Figure 1.** Expression of Munc13 isoforms in the mouse adrenal gland. KO mouse lines of the respective Munc13 isoform were used as control. The antibodies used to detect individual Munc13 isoforms and loading controls are indicated on the left. (**A**) Munc13-1 (*) is barely detectable in perinatal adrenal gland. (**B**) ubMunc13-2, but not bMunc13-2, is expressed. (**C**) Munc13-3 was not detected. (**D**) Baiap3 was detected, but not (**E**) Munc13-4. *Jx* refers to mice homozygous for the *Unc13d*[Jinx] mutation (**Crozat et al., 2007**). (**F**) Munc13-1 and Baiap3 are mainly located in the medulla (Med), but ubMunc13-2 is present in cortex (Cort) as well. Please note that the difference in the position of ubMunc13-2 relative to the marker in panels (**B**) and (**F**) is due to how far the respective gels were run. Loading controls were valosin-containing protein (VCP), glyceraldehyde-3-phosphate dehydrogenase (GAPDH) and CgA. Brain samples, and spleen tissue in the case of Munc13-4, were used for comparison. Please note that the *Baiap3*[KO] (**Wojcik et al., 2013**) and *Unc13a*[KO] animals express truncated protein products, whereas the truncated product present in the *Unc13b*[KO] (**Cooper et al., 2012**) is not shown here. Based on previous analyses of the *Unc13a*[KO] and *Unc13b*[KO] mice (**Augustin et al., 1999**; **Cooper et al., 2012**; **Varoqueaux et al., 2002**), the truncated Munc13-1 and Munc13-2 products are neither functional, nor do they have a dominant-negative effect. The truncated Baiap3 product was not detected in adrenal gland, and its effect in the brain, where it can be detected in young animals up to P21, is currently unknown (**Wojcik et al., 2013**). See also *Figure 1—figure supplement 1*.

The following figure supplement is available for figure 1:

**Figure supplement 1.** Comparison of Munc13-1, Munc13-2, and Munc13-3 expression.

supplement 1B), the ubiquitous isoform ubMunc13-2 (*Figure 1B* and *Figure 1—figure supplement 1B*), and Baiap3 (*Figure 1D*). Not detected were the brain-specific isoform of Munc13-2 (bMunc13-2), which is a splice variant expressed from the same gene as ubMunc13-2 (*Figure 1B*), Munc13-3 (*Figure 1C*), and the non-neuronal isoform Munc13-4 (*Figure 1E*). To directly compare the expression levels of Munc13-1, ubMunc13-2, bMunc13-2, and Munc13-3, we used knock-in mice that express these proteins fused to enhanced yellow or green fluorescent protein (EYFP/EGFP) from the respective endogenous loci (*Cooper et al., 2012*; *Kalla et al., 2006*). We found that ubMunc13-2-EYFP is the only isoform readily detectable in the adrenal gland using an antibody to the GFP-derived tags (*Figure 1—figure supplement 1A*).

To assess whether the isoforms detected in whole gland homogenates are present in the adrenal medulla, and/or the adrenal cortex, we dissected adult wild-type (WT) adrenal glands, and used an antibody to the LDCV marker Chromogranin A (CgA) to monitor effective separation of the medullary tissue, which consists mostly of chromaffin cells, from cortical tissue (*Figure 1F*). The expression

of Munc13-1 and Baiap3 in the adrenal gland is largely restricted to the medulla. Expression of ubMunc13-2 was detected in both adrenal medulla and cortex. Thus, a significant fraction of the ubMunc13-2 signal detected in whole gland homogenates (*Figure 1B*) appears to originate from the adrenal cortex, possibly due to innervation of the cortex by ubMunc13-2 positive synapses.

## Absence of Munc13-1, Munc13-3 or Baiap3 does not impair LDCV exocytosis in chromaffin cells

Next, we analyzed cultured chromaffin cells from knockout (KO) mice deficient for the individual Munc13 isoforms (*Figure 2*). LDCV exocytosis was triggered using flash photolysis of caged $Ca^{2+}$, which causes a sharp global increase in intracellular $[Ca^{2+}]$ (*Neher, 2006*). Fusion of LDCVs with the plasma membrane was monitored by measurement of the membrane capacitance change ($\triangle Cm$). Fitting a sum of three exponentials to the exocytotic burst of each individual trace identifies the amplitudes and time constants of release, which are generally interpreted as two kinetically distinct vesicle pools, the fast burst as the Readily-Releasable Pool (RRP), and the slow burst as the Slowly-Releasable Pool (SRP) (*Sorensen et al., 2003a*; *Voets, 2000*). However, as will be discussed later, the slow burst component may in fact not be a releasable pool, but instead represent the conversion from a Non-Releasable Pool (NRP), to the RRP (*Walter et al., 2013*). The rate of sustained release was measured as a linear component after the exocytotic burst, and reflects the ongoing recruitment of LDCVs into the NRP/SRP and RRP. Deletion of Munc13-1 (*Unc13a*), the major Munc13 isoform in SV exocytosis (*Augustin et al., 1999*; *Varoqueaux et al., 2002*), did not markedly alter LDCV exocytosis compared to WT littermate controls (*Figure 2A,D*), nor did it affect the kinetics of the exocytotic burst (*Figure 2D*).

Although we did not detect Munc13-3 in the adrenal gland, we wanted to rule out possible physiological effects of protein expression below the detection limit of Western blot analysis (*Figure 1C*), and included *Unc13c*[KO] mice in our analysis. However, as expected, LDCV exocytosis in *Unc13c*[KO] chromaffin cells was not perturbed (*Figure 2B,D*).

We then investigated the possible role of Baiap3 in LDCV exocytosis, as this isoform is prominently expressed in the adrenal medulla (*Figure 1D,F*). Surprisingly, LDCV exocytosis in *Baiap3*[KO] cells was intact (*Figure 2C,D*). Furthermore, *Baiap3*[KO] cells also did not show a release deficit when we stimulated the cells using a series of depolarization steps (*Figure 2—figure supplement 1A–C*), nor did overexpression of Baiap3 in WT cells affect LDCV exocytosis (*Figure 2—figure supplement 1D–F*).

## Absence of ubMunc13-2 dramatically reduces LDCV release

We then analyzed the role of ubMunc13-2 and Munc13-1 in chromaffin cell LDCV exocytosis. For this purpose, we used an *Unc13a/b* (DKO) mouse line. Heterozygous (Het) animals of this line express ~50% of WT levels of Munc13-1 and Munc13-2, which does not affect neurotransmission (*Augustin et al., 1999*; *Varoqueaux et al., 2002*). Data were collected from genotype groups available for a given litter and were pooled for analysis. Because our breeding scheme did not produce littermate WT animals in sufficient numbers, and because deletion of *Unc13a* alone was without effect, data from *Unc13a*[WT]*Unc13b*[Het] and *Unc13a*[Het]*Unc13b*[Het] cells were pooled and used as control (*Figure 2E–J*, *Unc13a*[WT/Het]*Unc13b*[Het]).

Deletion of both *Unc13a* alleles together with a single *Unc13b* allele (*Unc13a*[KO]Unc13b[Het]) did not reduce LDCV release (*Figure 2E,F*). By contrast, abrogation of ubMunc13-2 expression alone, irrespective of the *Unc13a* genotype, drastically diminished release (*Figure 2E,F*). Furthermore, in the context of the *Unc13b*[KO] background, cells with *Unc13a*[WT], *Unc13a*[Het], and *Unc13a*[KO] genotypes showed a progressive reduction of LDCV release that depended on the number of *Unc13a* alleles present (*Figure 2F,G*). The fast and slow burst components were reduced to 39%, 32%, and 27%, and to 54%, 52%, and 42% of control levels, respectively (*Figure 2F*). The rate of sustained release was reduced even more dramatically, to 26%, 19%, and 12% of control levels (*Figure 2F*). When one uses the *Unc13a*[WT]*Unc13b*[KO] genotype as a reference point (*Figure 2G*), the deletion of *Unc13a* caused a reduction of the sustained release component to 48%. The rate of sustained release of *Unc13a*[KO]*Unc13b*[KO] cells was also significantly reduced when compared to *Unc13a*[Het]*Unc13b*[KO] cells.

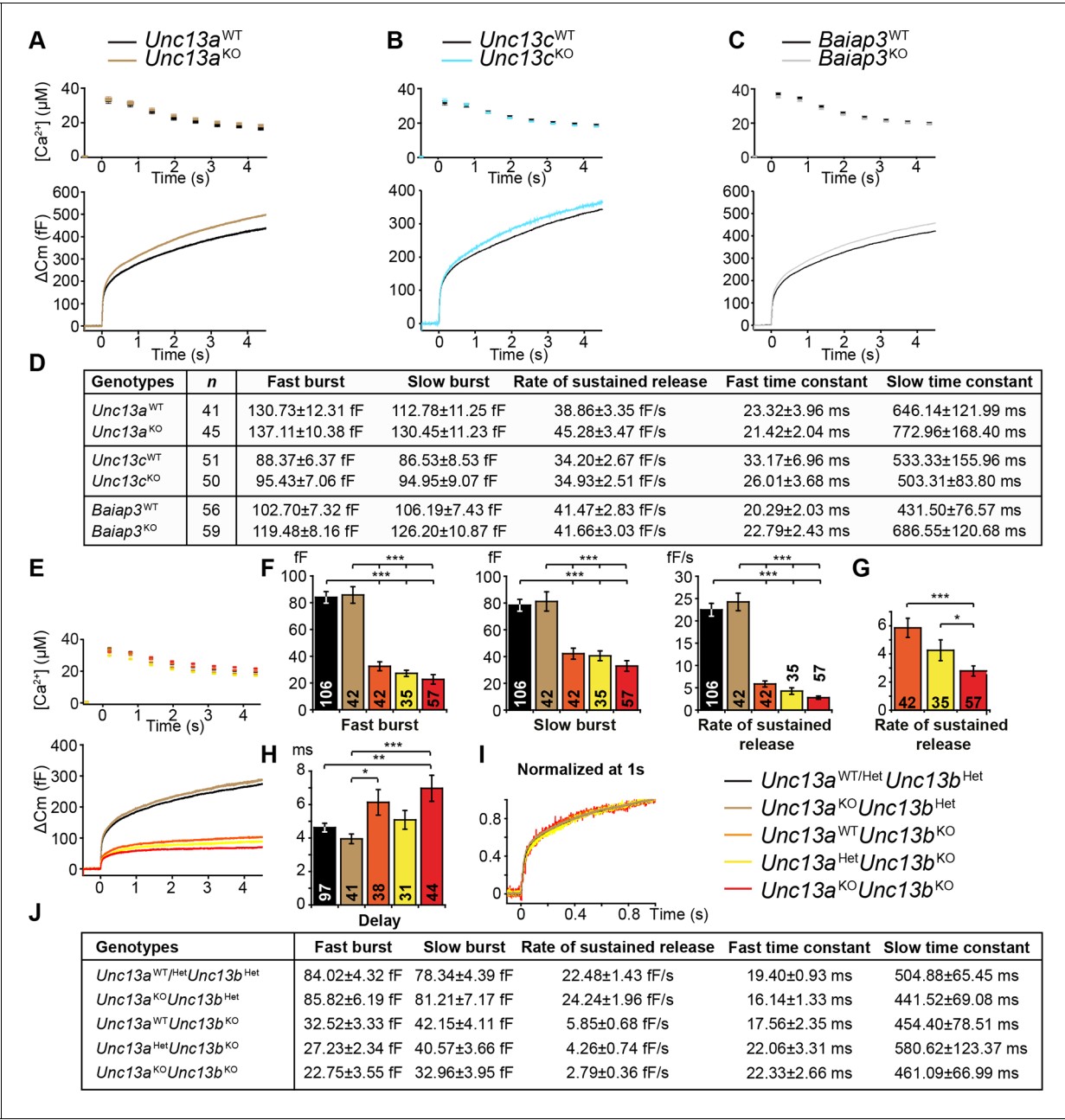

**Figure 2.** Flash photolysis induced LDCV exocytosis in chromaffin cells. For each KO line, the average intracellular $[Ca^{2+}]$ ± SEM and the average $\triangle Cm$ are shown in panels (A–C, E). Single gene deletions of (A) *Unc13a*, (B) *Unc13c*, or (C) *Baiap3* did not impair LDCV exocytosis. (D) Summary of burst sizes, sustained release rates, and time constants. (E) LDCV exocytosis is dramatically reduced in *Unc13a*[KO]*Unc13b*[KO] cells, *Unc13a*[WT]*Unc13b*[KO] cells, and *Unc13a*[Het]*Unc13b*[KO] cells. This reduction is primarily due to the absence of ubMunc13-2. (F) Fast burst, slow burst and the rate of sustained release are reduced in the absence of Munc13-1 and ubMunc13-2, as well as in the absence of ubMunc13-2 alone (ANOVA with post-hoc Tukey's test). (G) Compared to *Unc13a*[WT]*Unc13b*[KO] cells, the deletion of Munc13-1 causes significant reductions in the slow burst and the rate of sustained release (Student's t-test, two-tailed). (H) Delay of the onset of exocytosis after the flash stimulus (ANOVA with post-hoc Tukey's test). (I) Normalized traces show identical release kinetics of the exocytotic burst. (J) Summary of the release components shown in panels (E, F, G). Time constants are not significantly different (ANOVA with post-hoc Tukey's test). (*p < 0.05, **p < 0.01, ***p < 0.001). See also *Figure 2—figure supplement 1*.

The following figure supplement is available for figure 2:

**Figure supplement 1.** Single gene deletion or overexpression of Baiap3 does not affect LDCV exocytosis in chromaffin cells.

The deletion of both *Unc13a* and *Unc13b* significantly delayed the onset of vesicular exocytosis triggered by flash photolysis, compared to control and *Unc13a*[KO]*Unc13b*[Het] cells (*Figure 2H*). *Unc13a*[WT]*Unc13b*[KO] cells also showed a mild increase in delay. However, this difference was significant only when compared to *Unc13a*[KO]*Unc13b*[Het] cells, but not compared to the other groups.

Thus, ubMunc13-2, the only isoform expressed from the *Unc13b* gene in mouse chromaffin cells, is the most critical isoform for LDCV release in this cell type. Moreover, in its absence it becomes apparent that endogenous Munc13-1 also regulates LDCV release in this cell type.

## Reduced IRP and RRP in the absence of ubMunc13-2

We next assessed whether ubMunc13-2 affects LDCV release in response to $Ca^{2+}$ entry through voltage-gated $Ca^{2+}$ channels by stimulating the cells with a series of depolarization steps (*Figure 3*). The first six short depolarizations of the train release the Immediately-Releasable Pool (IRP), that is, the subset of RRP vesicles located closest to $Ca^{2+}$-channels (*Schonn et al., 2010*; *Voets et al., 1999*). We found a significant reduction in LDCV release; the size of the RRP in *Unc13b*[KO] cells was reduced to 53% of WT levels (*Figure 3C*). This deficit is somewhat less pronounced than the reduction seen in the flash photolysis experiment (reduction to 39%, *Figure 2F*), most likely because the depolarization protocol used to obtain the data shown in *Figure 3* lasts several seconds and therefore causes

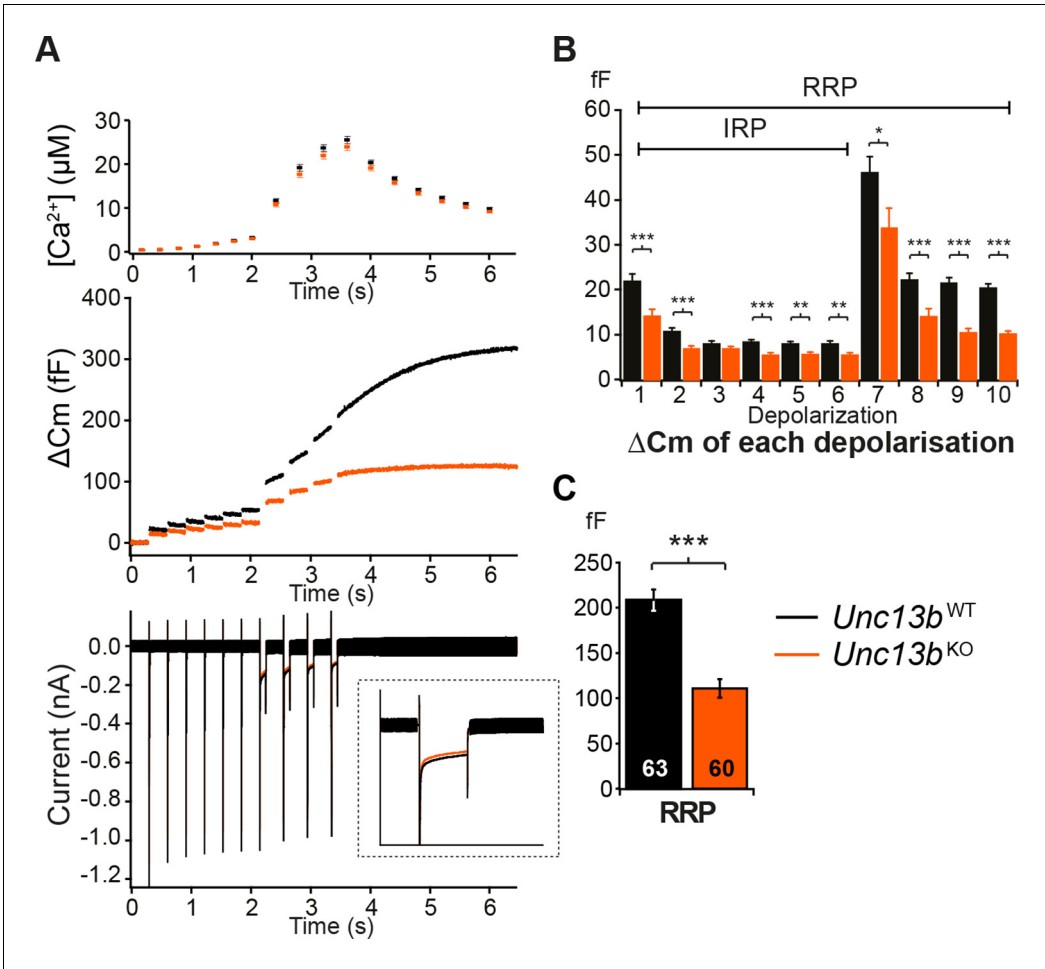

**Figure 3.** Absence of Munc13-2 results in a significant release deficit in response to depolarization. (**A**) Shown are the averaged $[Ca^{2+}] \pm$ SEM, △Cm, and whole-cell current traces of *Unc13b*[WT] and *Unc13b*[KO] cells. The inset in the whole-cell current panel shows an enlargement of the first 100 ms depolarization. (**B**) △Cm elicited by individual depolarizations was significantly different between the two groups. RRP, Readily-releasable Pool; IRP, Immediately-Releasable Pool. (**C**) The size of the RRP was measured as the △Cm after the train of depolarization pulses and was significantly reduced in *Unc13b*[KO] cells. (*p < 0.05, **p < 0.01, ***p < 0.001; Student's *t*-test, two-tailed).

some ongoing recovery of the RRP. By contrast, in flash photolysis experiments, the RRP is probed within ~60 ms (3 times the time constant), which is much faster than the recovery of the RRP. Strikingly, in the depolarization experiment, impaired release in $Unc13b^{KO}$ cells was already evident in response to the first 10-ms depolarization (*Figure 3B*), which implies that lack of ubMunc13-2 would even affect resting level catecholamine release driven by low frequency stimulation (*Zhou and Misler, 1995*).

## Reduced catecholamine release in the absence of ubMunc13-2

To understand how ubMunc13-2 affects the kinetics of single catecholamine release events, we performed single spike amperometry while infusing the cells with a solution with moderate (~4.6 µM) $[Ca^{2+}]$ (*Figure 4*). $Unc13b^{KO}$ cells showed a dramatic reduction in spike frequency (*Figure 4B,C*), whereas basic spike parameters such as duration, half-width, maximum amplitude, charge, rise time, and decay time were unchanged (*Figure 4D–I*). Amplitude, duration and charge of the spike foot signal, which is thought to reflect release during the initial formation of the fusion pore prior to full fusion, were also unchanged (*Figure 4J–L*). However, we found that the number of spikes that did show these foot signals was slightly reduced in $Unc13b^{KO}$ cells (*Figure 4M*), which may indicate that fusion pore dynamics are altered for some release events. However, overall, the LDCVs undergoing

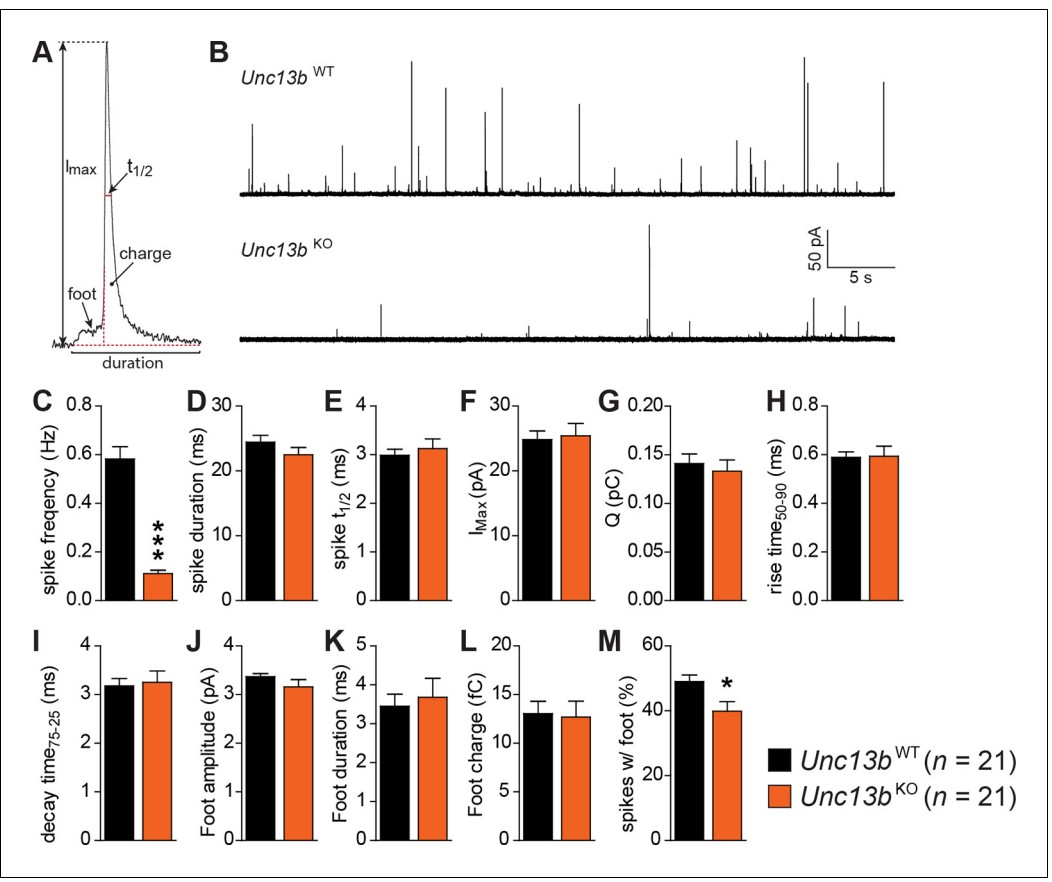

**Figure 4.** Reduced number of fusion events of catecholamine-containing LDCVs in the absence of Munc13-2. (**A**) Illustration of a single amperometric spike, corresponding to the release of catecholamines from a single LDCV, and the parameters analyzed. (**B**) Representative amperometric recordings of a $Unc13b^{WT}$ and a $Unc13b^{KO}$ cell. (**C**) Dramatic reduction in spike frequency in $Unc13b^{KO}$ chromaffin cells. (**D**) Spike duration, (**E**) width at half amplitude ($t_{½}$), (**F**) maximum spike amplitude, (**G**) amperometric charge, (**H**) rise time, and (**I**) decay time were unchanged in the $Unc13b^{KO}$. The stability of fusion pores was also not altered, as shown by the unchanged (**J**) foot amplitude, (**K**) duration and (**L**) charge. (**M**) The fraction of amperometric spikes with a detectable foot was reduced in $Unc13b^{KO}$ cells. (*p < 0.05, ***p < 0.001; Student's *t*-test, two-tailed).

fusion in the absence of ubMunc13-2 do so without major alterations in fusion kinetics or vesicle content.

## Munc13 isoforms display different LDCV priming efficiencies

As our experiments so far showed that, with the exception of Baiap3, the contribution of the Munc13 isoforms to the regulation of LDCV release correlates with their level of expression in peri-natal adrenal glands, we next wanted to compare the intrinsic properties of the different isoforms. To this end, we overexpressed Munc13-1, ubMunc13-2, Baiap3, and its closest relative Munc13-4 using Semliki Forest Virus (SFV) in *Unc13a*KO*Unc13b*KO cells. Munc13-1 and ubMunc13-2 were expressed as EGFP fusion constructs whose functions are identical to those of the respective WT proteins (*Rosenmund et al., 2002*), whereas Baiap3 and Munc13-4 were expressed as internal ribo-some entry site (IRES)-EGFP constructs, to avoid possible confounding effects of a fusion tag. *Unc13a*KO*Unc13b*KO cells expressing only EGFP were used as control. For the purpose of compari-son, the averaged traces obtained from the rescue experiments with the four isoforms were plotted in the same graph (*Figure 5A*). The exocytotic burst was measured as the △Cm within the first 0.5 s after the flash stimulus, and the rate of sustained release was measured as the △Cm between 0.5 s and 4 s after the flash (*Figure 5B*). Interestingly, Munc13-1 and ubMunc13-2 were both able to res-cue the LDCV release deficit of *Unc13a*KO*Unc13b*KO cells (*Figure 5A,B*). However, rescue with ubMunc13-2 resulted in an enormous enhancement of LDCV exocytosis to levels that by far exceeded the amount of exocytosis typical of WT cells, for both the exocytotic burst and the rate of sustained release (*Figure 5A,B*). The direct comparison of Munc13-1 and ubMunc13-2 expressing cells with matching EGFP fluorescence intensity confirmed that the stronger enhancement of burst size and rate of sustained release in ubMunc13-2 expressing cells was not due to higher expression levels of ubMunc13-2 (*Figure 5—figure supplement 1*).

Overexpression of Baiap3 failed to rescue the LDCV release deficit of the *Unc13a*KO*Unc13b*KO chromaffin cells (*Figure 5A,B*). Yet, its closest relative, Munc13-4, which regulates SNARE-mediated

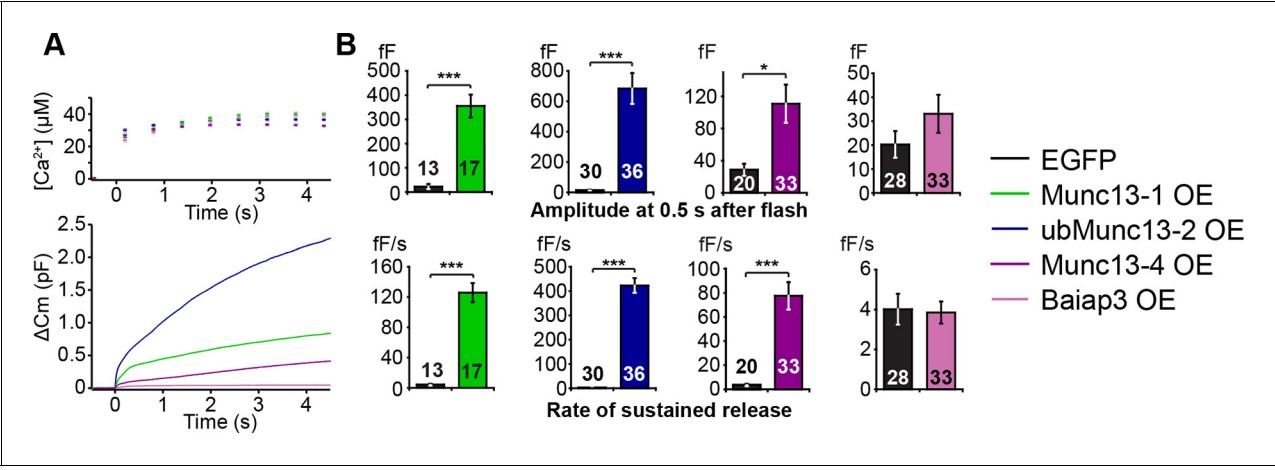

**Figure 5.** LDCV exocytosis in *Unc13a*KO*Unc13b*KO chromaffin cells is rescued by overexpression (OE) of Munc13-1, ubMunc13-2 and Munc13-4, but not Baiap3. *Unc13a*KO*Unc13b*KO cells were infected with SFV-Munc13-1-EGFP, SFV-ubMunc13-2-EGFP, SFV-Munc13-4-IRES-EGFP or SFV-Baiap3-IRES-EGFP using SFV-EGFP as control. (**A**) Averaged [Ca²⁺] ± SEM and capacitance traces △Cm are shown in the same graph to compare the efficiency of rescue: ubMunc13-2 > Munc13-1 > Munc13-4 > Baiap3. (**B**) Burst sizes and rates of sustained release after analysis of individual traces. (*p < 0.05, ***p < 0.001; Student's *t*-test, two-tailed). See *Figure 5—figure supplement 1* for a direct comparison of Munc13-1-EGFP- and ubMunc13-2-EGFP-expressing cell matched for fluorescence intensity, and *Figure 5—figure supplement 2* for western blotting to confirm the expression of Munc13-4 and Baiap3 from the IRES-constructs.

The following figure supplements are available for figure 5:

**Figure supplement 1.** Direct comparison of Munc13-1 and ubMunc13-2 over-expressing cells with matching EGFP fluorescence.

**Figure supplement 2.** Western blot analysis confirming expression of Munc13-4 and Baiap3 SFV constructs.

vesicular exocytosis in the hematopoietic system (*Feldmann et al., 2003*; *Shirakawa et al., 2004*), was able to rescue LDCV exocytosis in *Unc13a*[KO]*Unc13b*[KO] cells, albeit less efficiently than Munc13-1 or ubMunc13-2 (*Figure 5A,B*). To exclude the possibility that these findings might be due to inefficient translation of Baiap3 and Munc13-4, protein expression was confirmed with isoform-specific antibodies in SFV-infected neuronal cultures, which provide enough material for Western blot analysis (*Figure 5—figure supplement 2*). Thus, individual Munc13 isoforms appear to show inherent differences in their ability to promote LDCV release in chromaffin cells.

## LDCV docking in the absence of ubMunc13-2 and Munc13-1

We went on to investigate whether *Unc13a*[KO]*Unc13b*[KO] chromaffin cells show an LDCV docking defect analogous to the SV docking defect seen in *Unc13a*[KO]*Unc13b*[KO] synapses (*Siksou et al., 2009*). SV docking deficits of Munc13/Unc-13 deficient synapses in mice and *C. elegans* were previously only detected when rapid cryo-fixation methods were employed instead of classical chemical fixation for ultrastructural analysis (*Siksou et al., 2009*; *Weimer et al., 2006*). Moreover, it has been shown that 3D electron tomography (ET) allows a more accurate assessment of SV docking at the active zone (*Imig et al., 2014*; *Siksou et al., 2009*). To study LDCV recruitment and docking in chromaffin cells, we therefore combined high-pressure freezing (HPF) and freeze-substitution of acute adrenal gland slices with classical 2D-EM (*Figure 6A–E*) and high-resolution 3D-ET analyses (*Figure 6F–N*). Quantitative analysis of 2D-EM images of *Unc13a*[Het]*Unc13b*[Het] (*Figure 6A*) and *Unc13a*[KO]*Unc13b*[KO] chromaffin cells (*Figure 6B*) did not reveal any differences in LDCV distribution within 2 µm of the plasma membrane (PM) (*Figure 6C*), in the number of membrane-proximal LDCVs within 40 nm of the PM (*Figure 6D*), or in the total number of LDCVs (*Figure 6E*). LDCV docking and recruitment into the vicinity of the PM were assessed using 3D-ET (*Figure 6F–N*). From all LDCVs analyzed within 100 nm of the PM, the percentage of membrane-proximal LDCVs (0–40 nm) (*Figure 6M*) and their distribution (*Figure 6L*) was unaltered between both groups, indicating that LDCV recruitment to the PM is intact in Munc13-deficient chromaffin cells. The number of docked LDCVs, defined as LDCVs in physical contact with the PM and assigned to the 0–4 nm bin in *Figure 6L*, and the number of docked LDCVs normalized to the number of membrane-proximal LDCVs (*Figure 6N*) were unchanged. Furthermore, the average LDCV diameter of docked or non-docked LDCVs, measured by 3D-ET, did not differ significantly between genotypes, although *Unc13a*[KO]*Unc13b*[KO] LDCVs tended to be smaller (*Figure 6—figure supplement 1*). Thus, in spite of the dramatic release deficit seen in *Unc13a*[KO]*Unc13b*[KO] chromaffin cells and the dramatic SV docking deficit seen in neurons of this genotype (*Siksou et al., 2009*), we did not detect any changes in LDCV docking, nor a loss or accumulation of LDCVs in the vicinity of the PM. Thus, chromaffin cells can generate what appears to be a full-sized pool of morphologically docked LDCVs in the absence of Munc13-1 and Munc13-2, which implies that the molecular requirements of morphological LDCV and SV docking are distinct. Additionally, this could either indicate that the mechanism of functional docking, that is, priming, differs between LDCV and SVs as well, or else, that the primed LDCVs (i.e., those that belong to the RRP) are in the minority among the docked vesicles and therefore cannot be detected.

To distinguish between these two possibilities, we estimated the total number of docked LDCVs per cell. To this end, we re-calculated the percentage of docked vesicles identified using 3D-ET, as the percentage of membrane proximal vesicles (0–40 nm) identified in the 2D-EM analysis and converted LDCVs/µm PM to LDCVs/cell as described (*Parsons et al., 1995*). This conversion was necessary due to the limited volume sizes analyzed by 3D-ET and the uneven distribution of LDCVs within the cells. The estimated size of the morphologically docked pool was ~662 LDCVs per chromaffin cell in control cells. For *Unc13a*[KO]*Unc13b*[KO] cells we calculated ~865 docked LDCVs per cell, which can be accounted for by two factors used in the calculation: *Unc13a*[KO]*Unc13b*[KO] cells are slightly larger, and their LDCVs are slightly smaller (*Figure 6—figure supplement 1*). Both values are lower than the previously reported ~1607 for embryonic day (E)18 murine chromaffin cells (*de Wit, 2010*), presumably reflecting improved discrimination between docked and undocked vesicles by 3D-ET. With a diameter of a docked LDCV of ~170 nm (*Figure 6—figure supplement 1E*) and assuming a specific membrane capacitance of 1 µF/cm$^2$, this corresponds to a vesicular capacitance of 0.91 fF, in excellent agreement with recent electrophysiological measurements of 0.94 fF (*Pinheiro et al., 2014*). Thereby, the size of the RRP, which is <40 fF at resting [Ca$^{2+}$] (*Voets, 2000*), corresponds to <44 vesicles from the total of ~662 in control cells, indicating that even with 3D-ET, the RRP will be

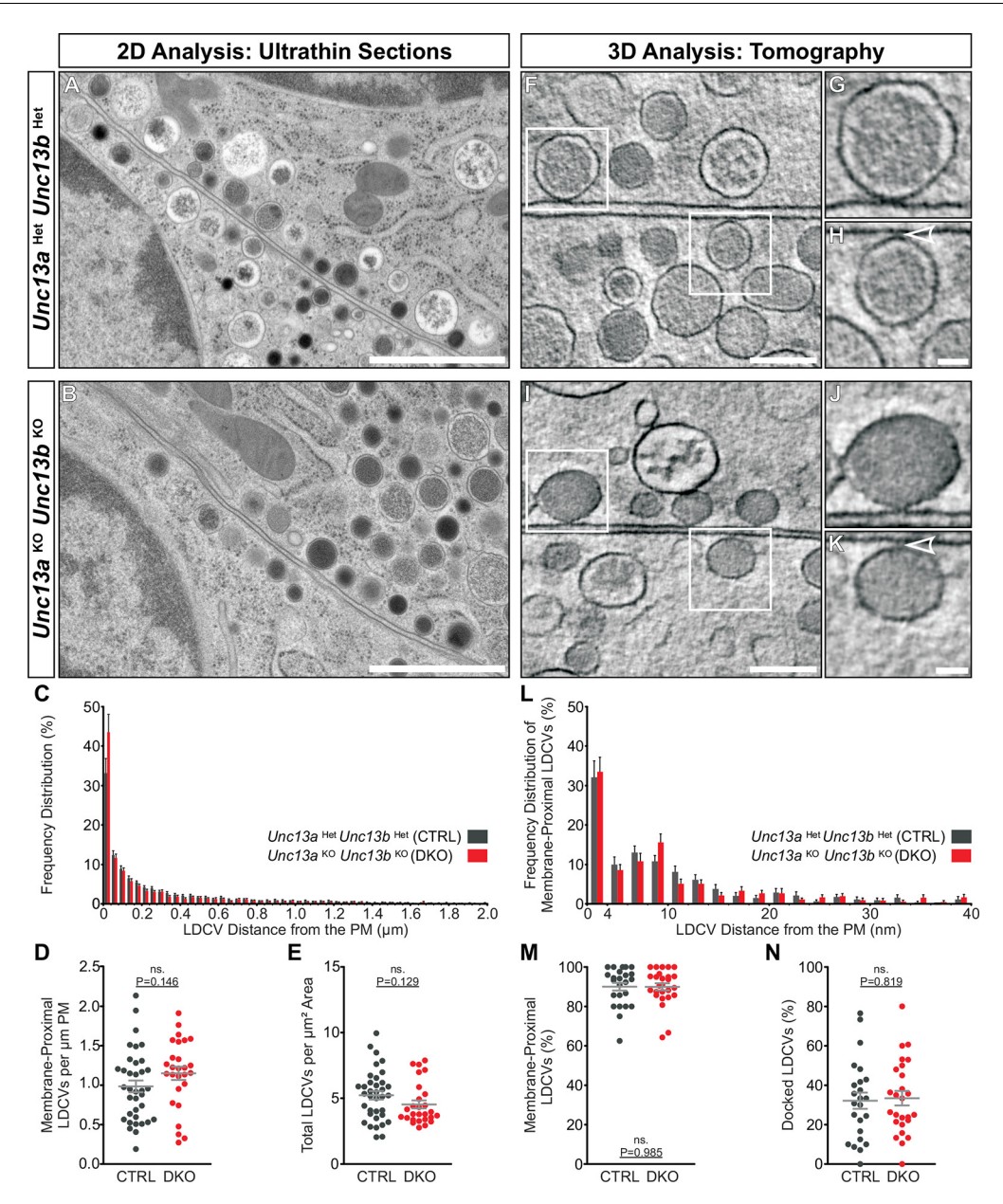

**Figure 6.** Ultrastructural analysis of LDCV docking in adrenal chromaffin cells. 2D-EM of (A) $Unc13a^{Het}Unc13b^{Het}$ (CTRL) and (B) $Unc13a^{KO}Unc13b^{KO}$ (DKO) adrenal glands. (C) Frequency distribution of LDCVs within 0–2 μm of the plasma membrane (PM). (D) Membrane-proximal LDCVs (0–40 nm of PM) normalized to PM circumference. (E) Total number of LDCVs normalized to cytoplasmic area. (F–K) Tomographically reconstructed subvolumes from 400 nm-thick sections through (F–H) $Unc13a^{Het}Unc13b^{Het}$ and (I–K) $Unc13a^{KO}Unc13b^{KO}$ cells in which docked LDCVs (enlarged in panels G,J) and undocked LDCVs (enlarged in panels H and K, small gaps separating undocked LDCVs from the PM indicated with arrowheads) can be distinguished. (L) Frequency distribution of membrane-proximal LDCVs distributed within 0–40 nm of the PM. (M) Number of membrane-proximal LDCVs expressed as a percentage of all LDCVs within 0–100 nm of the plasma membrane. (N) Percentage of docked LDCVs with respect to all membrane-proximal LDCVs (within 0–40 nm of PM). Scale bars represent 1 μm in (A,B), 200 nm in (F,I), and 50 nm in (H,K). *C: 5779 LDCVs in CTRL and 4120 LDCVs in DKO profiles. D, E: CTRL: N=2, n=36; DKO: N=2; n=27. L: 473 LDCVs in CTRL and 386 LDCVs in DKO tomographic subvolumes. M, N: CTRL: N=2, n=24;DKO: N=2, n=26.* Values indicate mean ± SEM. (Student's *t*-test, two-tailed). See also ***Figure 6—figure supplement 1***.

The following figure supplement is available for figure 6:

**Figure supplement 1.** Ultrastructural analysis of LDCV size in adrenal chromaffin cells.

very hard or even impossible to distinguish morphologically from other docked vesicles in adrenal chromaffin cells.

## Identification of a Munc13-sensitive step in LDCV priming

Thus far, our data indicate that although morphological docking of LDCVs does not require Munc13s, the priming of a functional RRP does. We therefore wanted to identify the Munc13-sensitive step in the LDCV priming process, and compare the intrinsic properties of Munc13-1 and ubMunc13-2, the two most relevant isoforms in the adrenal medulla. In a recently published mathematical model for LDCV fusion, we showed that the fast and slow bursts of release originate from two serially arranged pools of vesicles, the RRP and the non-releasable NRP, respectively (*Walter et al., 2013*) (*Figure 7*). The NRP in turn is refilled from a larger depot pool. Thus, the model features two separate priming steps (*Liu et al., 2010*), but only one fusion pathway (*Figure 7E*). Since the deletion of *Unc13a* and *Unc13b* changed the fast and slow burst to nearly the same degree (*Figure 2F*), within the framework of this model, Munc13s must act upstream of the NRP. Furthermore, since the sustained release rate is changed proportionally as well, Munc13s likely act to accelerate the forward priming rate, $k_1$ (*Ashery et al., 2000*). In most models, this rate constant $k_1$ is $Ca^{2+}$-dependent (*Voets, 2000*; *Walter et al., 2013*), and confers overall $Ca^{2+}$-dependence to the primed vesicle pool. We next investigated how the two relevant Munc13-isoforms in chromaffin cells (ubMunc13-2, Munc13-1) affect this priming step.

The $Ca^{2+}$-dependence of LDCV-priming (essentially $k_1$ in *Figure 7E*) can be assessed in an experiment by varying the pre-flash intracellular $[Ca^{2+}]$, before an uncaging flash is used to probe the size of the primed vesicle pool (*Voets, 2000*). We expressed either Munc13-1 or ubMunc13-2 in *Unc13a*-$^{KO}$*Unc13b*$^{KO}$ cells and extended the range of pre-flash $[Ca^{2+}]$ values from the previously used 300–600 nM (*Figure 2* and *Figure 5*) to 250–1200 nM (*Figure 7A* and *Figure 7—figure supplement 1*). In order to compare the respective $Ca^{2+}$-sensitivities rather than the absolute priming rates of Munc13-1 and ubMunc13-2 – and to overcome cell-to-cell variability – we normalized capacitance traces to their value after 3 s (*Figure 7A*, left and middle panels). Using the fractional increase in capacitance after 30 ms as a read-out of the primed vesicle pool, we identified the characteristic $Ca^{2+}$-dependence of priming. Strikingly, the $Ca^{2+}$-dependence was almost identical for the two isoforms and could be fitted with a single Hill equation (*Figure 7A* right-hand panel, *Table 1*).

Thus, $Ca^{2+}$-dependent priming is supported with identical steady-state affinities in the presence of Munc13-1 or ubMunc13-2. However, when applying $Ca^{2+}$-uncaging flashes from a relatively low pre-flash $[Ca^{2+}]$, the two isoforms induce quite different secretion kinetics (*Figure 7B,C*). For ubMunc13-2, secretion shows a clear sigmoid shape with acceleration after ~0.5 s (*Figure 7C*), which is absent when the pre-flash $[Ca^{2+}]$ is higher. This sigmoid shape of ubMunc13-2 driven secretion was noted before and was attributed to a slow association of ubMunc13-2 with Calmodulin and $Ca^{2+}$, resulting in a slow 'priming switch' (*Zikich et al., 2008*). In contrast, Munc13-1 does not show this secondary acceleration (*Figure 7B*), regardless of the pre-flash $[Ca^{2+}]$. To understand the origin of this behavior, we modeled the $Ca^{2+}$ association with the priming sensor (PS) explicitly – in our previous model (*Walter et al., 2013*), this step had been assumed to be always in equilibrium. In accordance with the observed identical steady state $Ca^{2+}$ dependencies (*Figure 7A*), we used identical dissociation constant ($K_D$) values for both isoforms, and varied only the on-rate, $k_{on}$, (the off-rate was changed simultaneously, $k_{off} = K_D*k_{on}$, with constant $K_D$). This led to a very satisfactory fit to both Munc13-1 and ubMunc13-2 data from both low and high pre-flash $[Ca^{2+}]$ (*Figure 7B,C*). Also, this model made it possible to fit both the control trace, and the *Unc13b*$^{KO}$ trace (*Figure 7D*). Importantly, the fit was performed simultaneously to all traces, so that we could ensure consistency between fits, simplify the interpretation of parameter changes, and ensure that all conditions could be reproduced by one version of our model (for fitted parameters, refer to *Table 1*).

Our model (*Figure 7E*) assumes that the NRP vesicles can only fuse after maturing to the RRP state (*Walter et al., 2013*). Earlier models assumed that the 'NRP pool' can fuse directly via an alternative pathway; in those cases, the corresponding pool was called 'Slowly Releasable' (SRP) (*Voets, 2000*). We note that our conclusion that Munc13-1/ubMunc13-2 exert their main effects upstream of both pools (NRP/SRP and RRP), i.e. on $k_1$, is consistent with both ideas (see also *Ashery et al., 2000*). Therefore, our observations here do not necessarily distinguish between the parallel pool model (SRP and RRP are both releasable) and the sequential pool model (only the RRP is releasable); but see (*Walter et al., 2013*) for data supporting the sequential pool model.

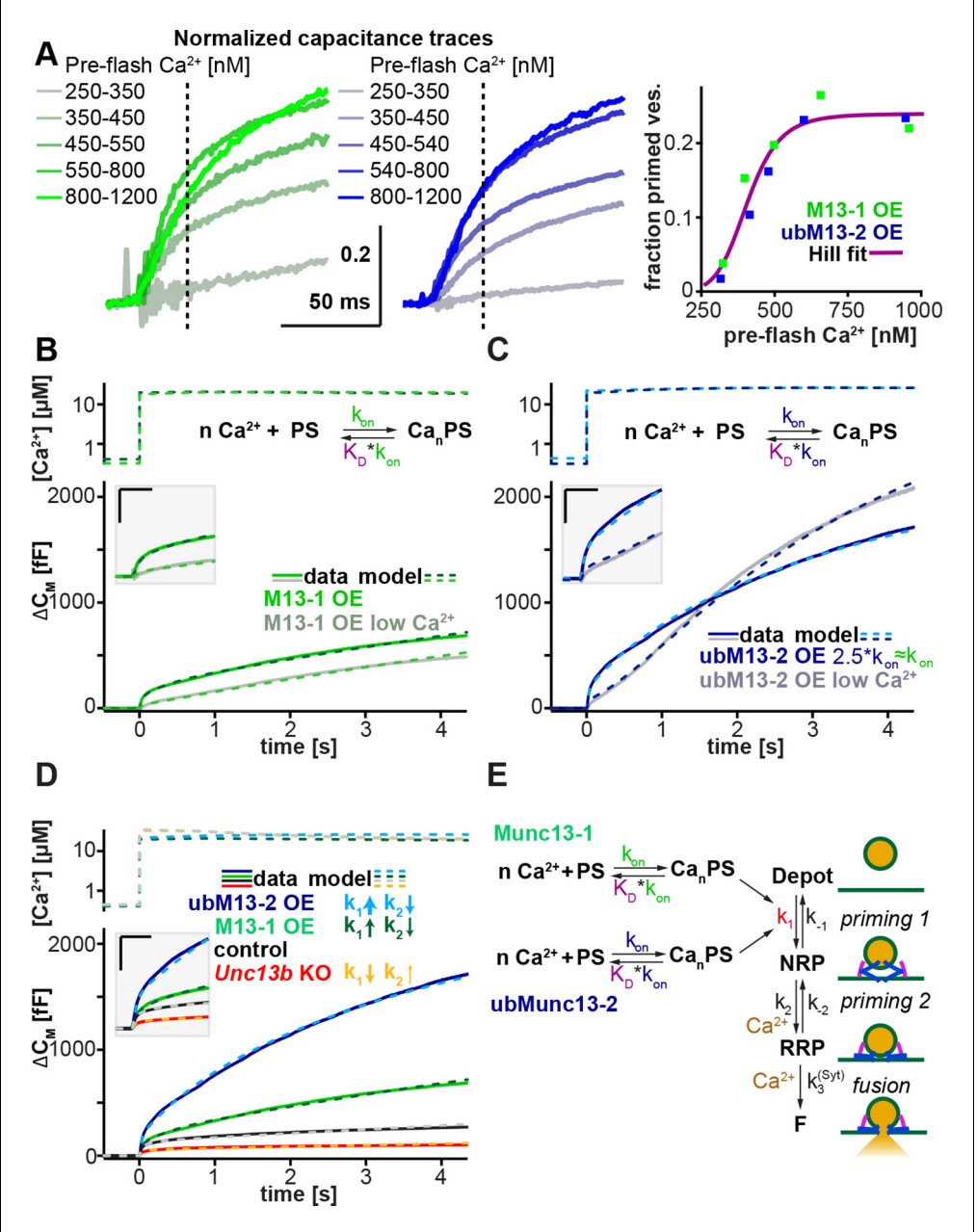

**Figure 7.** Munc13-1 and ubMunc13-2 accelerate upstream vesicle priming ('priming step 1') with identical $Ca^{2+}$ affinities, but distinct rates. (**A**) Estimation of steady-state $Ca^{2+}$ affinities of vesicle priming driven by Munc13-1 or ubMunc13-2. Left two panels: binned and averaged secretory responses in $Unc13a^{KO}Unc13b^{KO}$ cells overexpressing (OE) either Munc13-1 (green) or ubMunc13-2 (blue). The release fraction at 30 ms (traces normalized to their amplitude after 3 s) after the stimulus was determined as the read-out for priming (vertical broken lines). Right panel: the fraction of release plotted as a function of pre-flash $[Ca^{2+}]$. Both genotypes are described by the same Hill function, suggesting a similar $Ca^{2+}$-dependence of priming (i.e. identical cooperativity and affinity). (**B**) Fits of a secretion model (see panel **E**) to the capacitance responses observed experimentally in $Unc13a^{KO}Unc13b^{KO}$ cells expressing Munc13-1 at intermediate and low pre-flash $[Ca^{2+}]$. Top panel: measured $Ca^{2+}$ values were used to drive the secretion model. The chemical equation shows a priming sensor (PS), which is active in the $Ca^{2+}$ bound state ($Ca_nPS$). The lower panel shows the experimental capacitance data (solid lines) together with simulations with the best fit parameters (broken lines; see **Table 1**). Insert: magnified view with horizontal/vertical scale bars: 200 ms/200 fF. (**C**) Same as (**B**), but for $Unc13a^{KO}Unc13b^{KO}$ cells expressing ubMunc13-2. The secretion model was fitted using identical $K_D$s and n for the $Ca^{2+}$ binding to the PS (determined by the analysis shown in panel **A**). The best-fit values suggest a ~2.5-fold slower on-rate (activation rate), but a 4.5-

*Figure 7 continued*

fold higher maximal priming rate for ubMunc13-2 (see **Table 1**), resulting in a sigmoidal secretion response from low pre-flash [$Ca^{2+}$]. (**D**) Fitting our secretion model to the experimental data of several genotypes suggests that Munc13-1 and ubMunc13-2 both primarily act by increasing the forward priming rate ($k_1$, see also panel **E**), while loss of ubMunc13-2 – the dominant endogenous isoform – has the opposite effect. The downstream priming step ('priming 2', $k_2$) changes in the opposite direction. (**E**) Secretion model: Munc13-1 (green) and ubMunc13-2 (blue) regulate the $Ca^{2+}$ binding rates ($k_{on}$) to a PS, which controls the asymptotic forward priming rate $k_1$ (see text for details, **Table 1** for fitted parameters, and **Walter et al., 2013** for model development). NRP: Non-Releasable Pool; RRP: Readily-Releasable Pool; F: Fused pool. See also **Figure 7—figure supplement 1**.

The following figure supplement is available for figure 7:

**Figure supplement 1.** Exocytotic burst size as a function of pre-flash [$Ca^{2+}$].

---

The modeling resulted in two main conclusions (**Table 1**): First, as expected, the action of Munc13 (either isoform) is consistent with an increase in $k_1$ – the forward rate of priming within the first priming step (**Figure 7E**). This is seen both by the increase in the fitted $k_1$ upon overexpression of either isoform, and by the decrease of $k_1$ in the *Unc13b*[KO] cells. Second, ubMunc13-2 increases $k_1$ 4.5-fold more than Munc13-1, but it does so after a longer delay. In the model, this delay is due to slower kinetics of $Ca^{2+}$ binding to the priming sensor (**Zikich et al., 2008**). Note that this step might coincide with the translocation of Munc13-1 or ubMunc13-2 to the membrane as a prerequisite for the priming action of the protein. Thus, the different delays might reflect differences in the

**Table 1.** Parameters of the exocytosis model.

| Parameter | Control | *Unc13b*[KO] | Munc13-1 OE | ubMunc13-2 OE | Comment |
|---|---|---|---|---|---|
| $V_{tot}$ | 2350 | | | | Total number of vesicles, best fit |
| $k_1$ | $p(Ca^{2+}) \cdot k_{1max}$ | | | | |
| $p(Ca^{2+})$ | $\frac{(Ca^{2+})^n}{(Ca^{2+})^n+(K_{D,cat})^n}$ | | | | fraction of activated PS |
| $k_{on}$ | $8.96 \times 10^{-9}$ s$^{-1}$ µM$^{-n}$ | $2.25 \times 10^{-8}$ s$^{-1}$ µM$^{-n}$ | $2.25 \times 10^{-8}$ s$^{-1}$ µM$^{-n}$ | $8.96 \times 10^{-9}$ s$^{-1}$ µM$^{-n}$ | On-rate calcium binding to PS |
| $k_{1Max}$ | $1.99 \times 10^{-2}$ s$^{-1}$ | $6.89 \times 10^{-3}$ s$^{-1}$ | $7.44 \times 10^{-2}$ s$^{-1}$ | $3.42 \times 10^{-1}$ s$^{-1}$ | Maximal priming rate, best fit |
| $\sqrt[n]{K_D}$ | 0.407 µM | | | | Experiment, Hill plot Figure 7 |
| $k_{-1}$ | $4.70 \times 10^{-1}$ s$^{-1}$ | | | | Best fit |
| $n$ | 7.38 | | | | Cooperativity PS, experiment, Hill plot Figure 7 |
| $k_2$ | $k_{20} + g(Ca^{2+}) \cdot k_{2cat}$ | | | | (**Walter et al., 2013**) |
| $k_{-2}$ | $k_{-20} + g[Ca^{2+}] \cdot k_{-2cat}$ | | | | (**Walter et al., 2013**) |
| $g(Ca^{2+})$ | $\frac{Ca^{2+}}{Ca^{2+}+K_{D,cat}}$ | | | | (**Walter et al., 2013**) |
| $k_{20}$ | $2.37 \times 10^{-2}$ s$^{-1}$ | $2.95 \times 10^{-2}$ s$^{-1}$ | $1.29 \times 10^{-2}$ s$^{-1}$ | $5.80 \times 10^{-3}$ s$^{-1}$ | Best fit |
| $k_{2cat}$ | $3.95 \times 10^{1}$ s$^{-1}$ | $4.91 \times 10^{1}$ s$^{-1}$ | $2.14 \times 10^{1}$ s$^{-1}$ | $9.65 \times 10^{0}$ s$^{-1}$ | Best fit |
| $k_{-20}$ | $2.10 \times 10^{-2}$ s$^{-1}$ | | | | Best fit |
| $k_{-2cat}$ | $= k_{2cat} \cdot k_{-20}/k_{20}$ | | | | (**Walter et al., 2013**) |
| $K_{D, cat}$ | 138 µM | | | | Best fit |
| $k_3$ | 4.4 s$^{-1}$µM$^{-1}$ | | | | (**Voets, 2000**) |
| $k_{-3}$ | 56 s$^{-1}$ | | | | (**Voets, 2000**) |
| $k_4$ | 1450 s$^{-1}$ | | | | (**Voets, 2000**) |

membrane translocation step. As a minor note, we also noticed that $k_2$, the forward rate of down-stream priming ('priming 2', *Figure 7E*) always changed in the opposite direction of $k_1$. The reason for this is unclear, but one explanation could be that the protein(s) driving downstream priming compete with Munc13 for association with the fusion machinery.

## Discussion

Our study provides a comprehensive analysis of the Munc13 protein family in LDCV docking and priming, and shows that genetic deletion of Munc13-1 (*Unc13a*) and Munc13-2 (*Unc13b*) severely impairs LDCV release in chromaffin cells. Yet surprisingly, LDCV docking, unlike SV docking, does not require Munc13s. We furthermore identify the step most sensitive to Munc13s in the LDCV priming process, and show that ubMunc13-2 and Munc13-1 accelerate this step with identical $Ca^{2+}$ affinities but distinct $Ca^{2+}$ binding rates.

The essential role of Munc13s in vesicular release appears to lie in the opening of Syntaxin (Stx-1)/Munc18-1 complexes, to permit the formation of Stx-1/SNAP-25 heterodimers that act as docking platforms for the vesicular SNARE protein Synaptobrevin-2 (Syb-2) (*Hammarlund et al., 2007*; *Ma et al., 2011*; *Ma et al., 2013*; *Richmond et al., 1999*; *Sassa et al., 1999*; *Yang et al., 2015*). SNARE complex assembly is thought to proceed in an N- to C-terminal zipper-like fashion (*Fasshauer and Margittai, 2004*; *Pobbati et al., 2006*; *Sorensen et al., 2006*; *Walter et al., 2010*), and, at least for SVs, this assembly seems to be the molecular correlate of both the physical docking process, and acquisition of fusion competence, which is referred to as priming (*Imig et al., 2014*). However, as will be discussed below, morphological docking of LDCVs does not require Munc13s (*Figure 6*), yet the priming of a full-sized RRP does (*Figure 2*), indicating that LDCV docking and functional priming do not represent a one-step process in neuroendocrine cells.

### LDCV and SV docking have distinct requirements

Our data demonstrate that although Munc13s are critical for functional priming of LDCVs in chromaffin cells, morphological LDCV docking, even when assessed by 3D-ET at unprecedented resolution, is not impaired in the absence of Munc13s (*Figure 6*). Thus, in contrast to synapses, where most, if not all docked SVs are part of the RRP, the majority of docked LDCVs in chromaffin cells are not primed, and the functional RRP therefore cannot be distinguished from other docked LDCVs by current ultrastructural methods. Although we cannot completely exclude the possibility that Munc13-3 and Munc13-4 may be present at very low levels, that we were unable to detect (*Figure 1*), it seems unlikely that their presence could account for the full-sized pool of docked LDCVs in *Unc13a*-KO*Unc13b*KO cells.

This raises the question of how the non-primed LDCVs are docked. SV docking requires the SNARE proteins Stx-1, SNAP-25, and Syb-2, as well as Munc13s and CAPSs, but not necessarily the $Ca^{2+}$ sensor of fusion, Synaptotagmin-1 (Syt-1) (*Imig et al., 2014*). By contrast, current models imply that LDCV docking is mediated by Syt-1, possibly via interaction with the Stx-1/SNAP-25 acceptor complex (*de Wit et al., 2006*; *de Wit et al., 2009*; *Parisotto et al., 2012*). Additionally, Munc18-1 docks LDCVs via its interaction with the closed form of Stx-1 (*Gandasi and Barg, 2014*; *Gerber et al., 2008*; *Gulyas-Kovacs et al., 2007*; *Han et al., 2011*; *Voets et al., 2001*) and is also involved in an additional tethering step (*Toonen et al., 2006*). The vesicular SNAREs seem to be dispensable for docking in chromaffin cells (*Borisovska et al., 2005*; *Gerber et al., 2008*), although they have been implicated in PC12 cells (*Wu et al., 2012*). Some of these discrepancies are most likely due to methodological and terminological differences as well as to limitations in assessing true membrane attachment. However, since we used the same experimental approach previously employed to detect SV docking deficits (*Imig et al., 2014*; *Siksou et al., 2009*), our data clearly show that the molecular requirements of SV and LDCV docking are distinct. More specifically, while the formation of the docked/primed RRP requires Munc13s in both cases, and thus appears to be mechanistically quite similar for SVs and LDCVs, the non-primed LDCVs in chromaffin cells appear to dock via a separate, Munc13-independent mechanism.

Our findings are therefore consistent with the following model of LDCV docking and priming: (i) LDCV docking mediated by Munc18-1/Stx-1, this configuration would be the starting point for Munc13-mediated SNARE complex assembly, i.e. priming (*Ma et al., 2011*; *Ma et al., 2013*), and in parallel, (ii) LDCV docking mediated by a second configuration, that would not be expected to

progress to SNARE complex assembly directly or as efficiently, and thus be consistent with the large unprimed, but docked LDCV pool. What this second configuration would look like in terms of molecular interactions is less clear. Docking via Syt-1/Stx-1/SNAP-25 complexes would be consistent with un-primed docking (*de Wit et al., 2009*). This mode of docking would require the assumption that in chromaffin cells, Stx-1/SNAP-25 complexes can escape NSF/SNAP mediated disassembly. An additional or alternative mode of un-primed docking may involve the recruitment of vesicles based on the interaction of Syt-1 with phosphatidylinositol 4,5-bisphosphate (PIP2)/Stx-1 clusters (*Honigmann et al., 2013*; *Park et al., 2015*), although further interactions may be required to achieve close membrane apposition.

In the model suggested above, we explicitly included only molecular components for which docking deficits have been demonstrated in chromaffin cells. However, LDCV docking most likely involves additional factors. For instance, the docking of LDCVs to Munc18-1/Stx-1 complexes probably requires the interaction between Munc18 and the vesicle-associated small GTPases Rab3 and Rab27 (*Graham et al., 2008*; *Tsuboi and Fukuda, 2006*; *van Weering et al., 2007*), and additional docking/tethering factors may be involved in docking both primed and un-primed LDCVs to the membrane.

## Release kinetics of LDCV pools as correlates of SNARE complex assembly

Our analysis of how Munc13s prime LDCVs for fusion identifies the earliest phase of the priming process as the step most sensitive to Munc13s. We interpret our data according to a model that features Munc13 as a $Ca^{2+}$-sensitive priming protein in a single pathway to LDCV fusion with two serially arranged vesicle states or pools (NRP and RRP) (*Figure 7E*). This essentially allows us to describe what was previously interpreted as the release of two kinetically distinct LDCV pools (SRP and RRP), as two sequential priming processes, priming 1 and priming 2, resulting in only one releasable pool, the RRP (*Walter et al., 2013*). According to previous data, the step most sensitive to Munc13s – priming 1 – is also the step affected by mutations designed to interfere with the initiation of N-terminal SNARE complex assembly (*Walter et al., 2013*; *Walter et al., 2010*). This is in line with a function of Munc13s in initiating SNARE-complex assembly (*Yang et al., 2015*). The second priming step may involve a downstream, presumably more C-terminal phase of SNARE-complex assembly, although other options remain open. Thus, in the model (*Figure 7E*), the formation of the NRP, i.e. the step most sensitive to Munc13s, most likely represents the initiation of N-terminal SNARE complex assembly.

## Catalysis of fast and slow LDCV priming by Munc13-1 and ubMunc13-2

As our study demonstrates, Munc13 isoforms differ in their ability to facilitate LDCV priming (*Figure 5*). We detected three endogenously expressed isoforms in murine chromaffin cells, Munc13-1, ubMunc13-2, and Baiap3 (*Figure 1*). Baiap3, somewhat surprisingly given its prominent expression and ability to translocate to membranes in a $Ca^{2+}$-dependent manner (*Lecat et al., 2015*), does not appear to be involved in LDCV priming in this cell type. However, Munc13-4, which regulates SNARE- mediated vesicle exocytosis in the hematopoietic system (*Boswell et al., 2012*; *Feldmann et al., 2003*; *Shirakawa et al., 2004*), and is the closest relative of Baiap3 (*Koch et al., 2000*), can promote LDCV priming, albeit less efficiently than Munc13-1 and ubMunc13-2.

The two most relevant isoforms, Munc13-1 and ubMunc13-2, promote LDCV priming with very similar steady-state $Ca^{2+}$-affinities, but nonetheless confer unique release kinetics depending on the pre-stimulus $[Ca^{2+}]$. Modeling of the secretion kinetics produced by overexpression of Munc13-1 and ubMunc13-2 in *Unc13a*[KO]*Unc13b*[KO] cells allowed us to isolate the intrinsic properties of each isoform (*Table 1*). Secretion driven solely by the dominant isoform ubMunc13-2 shows a characteristic sigmoid shape at low pre-stimulus $[Ca^{2+}]$ (*Figure 7C*) (*Zikich et al., 2008*). In our secretion model, the best-fit parameters indicate 2.5-fold slower sensing of $[Ca^{2+}]$ for ubMunc13-2, which can however accelerate priming dramatically when $[Ca^{2+}]$ increases. Thus, the fitted maximum priming rate ($k_1$) for ubMunc13-2 is 4.5-fold higher than for Munc13-1. However, although Munc13-1 is unable to support the same maximum priming rate as ubMunc13-2, it reacts faster to a change in the $[Ca^{2+}]$ concentration (*Figure 7B*), which may reflect distinct conformational changes in response to $[Ca^{2+}]$, and/or differences in a membrane translocation step.

Thus, neuroendocrine cells can fundamentally modify the kinetics of secretion by expressing different Munc13 isoforms. Previous data from autaptic neurons showed that Munc13-1 causes short term depression, whereas ubMunc13-2 causes short-term facilitation (*Rosenmund et al., 2002*), which parallels our findings in chromaffin cells from low basal [$Ca^{2+}$], raising the possibility that the functions of different Munc13 isoforms in priming LDCVs and SVs are conserved, even though their role in docking is not. Our model therefore provides a theoretical framework for how the molecular properties of priming factors may be linked to the kinetics of exocytosis.

## Sequential actions of upstream and downstream priming catalysts

Although Munc13s have the strongest effect on the priming step 1, i.e. the formation of the NRP vesicle state, they also influence priming step 2, i.e. the formation of the RRP. Remarkably, overexpression and deletion of Munc13s change the rate constants $k_1$ (priming 1) and $k_2$ (priming 2) in opposite directions (*Figure 7D*). One possible reason for this effect could be that the interaction of Munc13s with the SNARE fusion machinery may compete with that of another priming protein, which mainly catalyzes priming step 2. A likely candidate for this second catalyst appears to be CAPS, as deletion of CAPS1 and CAPS2 leads to a significant reduction of the RRP, but has little effect on the NRP/SRP, placing Munc13 upstream of CAPS (*Liu et al., 2010*; *Liu et al., 2008*).

Furthermore, in PC12 cells, strong stimulation bypasses the need for CAPS-1 in LDCV exocytosis, but not the need for ubMunc13-2 (*Kabachinski et al., 2014*), and the ability of CAPS to promote membrane fusion is impaired by C-terminal mutations in Stx-1 (*Daily et al., 2010*). We therefore propose an LDCV priming model, in which the Munc13-driven priming step 1 corresponds to the initiation of N-terminal SNARE-complex assembly, and the CAPS-driven priming step 2 represents a more C-terminal, and presumably more easily completed phase of zippering (*Gao et al., 2012*). Assuming that priming step 2 is not catalyzed by CAPS alone but also influenced by Munc13 and possibly Syt-1, such a model would also offer an explanation as to why in both SV and LDCV exocytosis, lack of CAPS can be compensated for by an increase in $Ca^{2+}$, whereas lack of Munc13 cannot (*Jockusch et al., 2007*; *Kabachinski et al., 2014*).

## Conclusion

In summary, our data show that mammalian neurons and neuroendocrine cells both require Munc13s to generate fusion-competent vesicles, although the molecular steps leading to LDCV docking prior to SNARE complex assembly appear to be unique. In LDCV priming, the step most sensitive to Munc13s is the initial phase (priming step 1), which most likely corresponds to the initiation of N-terminal SNARE-complex assembly. Individual Munc13 isoforms accelerate this step at distinct rates, thereby imparting distinct properties on the kinetics of LDCV release, which indicates that they may have specialized functions in the fine-tuning of catecholamine release in response to varying physiological stimuli.

# Materials and methods

## Animals

All experiments were performed in compliance with the regulations of the local Animal Care and Use Committee of Lower Saxony, Oldenburg, Germany. The generation and basic characterization of the KO lines of the Munc13 isoforms has been described previously (*Augustin et al., 2001*; *Augustin et al., 1999*; *Varoqueaux et al., 2002*; *Wojcik et al., 2013*). *Unc13d*[KO] (*Unc13d*[Jinx]) mice (*Crozat et al., 2007*) were obtained from Jackson Laboratories. Adult and perinatal mice were killed by decapitation prior to the removal of adrenal glands and other tissues.

## Western blotting and antibodies

Adrenal glands of perinatal animals were excised and stored at -80°C prior to use. Adrenal glands from around 20 perinatal animals were pooled for the preparation of homogenates. Homogenates of whole adrenal glands were prepared by homogenization in an ice-cold buffer (320 mM D-glucose, 20 mM HEPES, 2 mM EDTA, pH 7.4, with 0.5 µg/ml leupeptin, 1 µg/ml aprotinin and 0.1 mM PMSF added freshly prior to homogenization), using a Potter S homogenizer. For the preparation of adrenal cortical and medullary homogenates, adrenal glands from WT animals were dissected in ice-cold

buffer containing 19 mM NaH$_2$PO$_4$ and 81 mM Na$_2$HPO$_4$ and material from 5–6 animals was pooled. Spleen homogenates were prepared similarly and the DNA in the samples subsequently digested with 0.66 U/μl benzonase (E1014, Sigma-Aldrich) in 3.86 mM MgCl$_2$ for 10 min at 37°C prior to denaturation. Whole-brain homogenates were prepared using a Potter S homogenizer and centrifuged for 10 min at 1000 *g* at 4°C to remove the nuclear fraction. Homogenates were analyzed by western blotting with the following antibodies at the indicated dilutions: rabbit-anti-Munc13-1 (1:500) (126103, Synaptic Systems), rabbit-anti-ubMunc13-2 (1:2000), rabbit-anti-bMunc13-2 (1:1000), rabbit-anti-Munc13-3 (1:500) (*Varoqueaux et al., 2005*), rabbit-anti-Baiap3 (1:1000) (*Wojcik et al., 2013*), goat-anti-Munc13-4 (1:250) (NB100-41385; Novus Biologicals), rabbit-anti-Chromogranin A (1:8000) (259002; Synaptic Systems), mouse-anti-GFP (1:500) (11814460001; Roche), mouse-anti-valosin containing protein (VCP) (1:1000) (612182; BD Transduction Laboratories), and mouse-anti-GAPDH (1:25000) (ab8245; Abcam). Secondary antibodies (goat anti-rabbit IgG, 111035144; goat anti-mouse IgG, 115035146, donkey anti-goat, 705-035-147) were obtained from Jackson ImmunoResearch.

## Chromaffin cell culture

Chromaffin cell cultures were prepared as described in Sørensen et al. (*Sorensen et al., 2003b*). Cultures from *Unc13a*$^{KO}$ and *Unc13a*$^{KO}$*Unc13b*$^{KO}$ mice were prepared on embryonic day (E)18, and from *Unc13b*$^{KO}$, *Unc13c*$^{KO}$, and *Baiap3*$^{KO}$ mice on postnatal day (P)0, in each case using littermates of the appropriate genotypes as controls. For overexpression of Baiap3 in WT cells, chromaffin cells from P0 WT C57Bl/6N mice were used. Briefly, adrenal glands were excised and placed into ice-cold Locke's solution (154 mM NaCl, 5.6 mM KCl, 0.84 mM NaH$_2$PO$_4$, 2.14 mM Na$_2$HPO$_4$ and 10 mM D-glucose, pH 7.0). The glands were then transferred to 300 μl of a papain solution [20 U/ml papain (Worthington Biochemical), 200 mg/L L-cysteine, 1 mM CaCl$_2$, 0.5 mM EDTA, in DMEM (Gibco)], which had been equilibrated for 15 min with 95% O$_2$ and 5% CO$_2$, and incubated with gentle shaking for 45 min at 37°C. To terminate the papain digestion, 300 μl of inactivating solution [10% fetal bovine serum (Gibco), 2.5 g/L trypsin inhibitor (Gibco) and 2.5 g/L albumin in DMEM (Gibco)] were then added, followed by an incubation period of 5 min at 37°C. The mixture of solutions was then replaced by 160 μl of DMEM (Linaris) supplemented with 1% insulin-transferrin-selenium X (Gibco) and 200 U/L penicillin-streptomycin (Gibco). The glands were triturated with a 200-μl pipette tip and the cell suspension was placed as 50 μl drops on coverslips in a 6-well plate. Following an incubation period of 30 min at 37°C at 8% CO$_2$ to allow cells to settle, 2 ml of DMEM (Linaris) with the supplements described above were added per well and the cells were kept at 37°C and 8% CO$_2$. The cells were used for electrophysiological recordings on days in vitro 2–3.

## Viral constructs

Expression constructs based on the SFV plasmid (pSFV1) for Munc13-1 and ubMunc13-2, both subcloned in frame with a C-terminal EGFP, have been described previously (*Rosenmund et al., 2002*). Munc13-4 and Baiap3 pSFV1 expression constructs were generated as IRES-EGFP constructs using the full-length cDNAs. Production of SFV particles was done according to published protocols (*Ashery et al., 1999*). Briefly, pSFV1 constructs and pSFV-helper2 DNA were linearized with Spe I and transcribed into RNA using SP6 RNA polymerase. RNA from the pSFV1 constructs and the pSFV-helper2 construct, 10 μg each, were electroporated (500 V, 0.957 mF) into baby hamster kidney 21 cells. Supernatant of cell cultures containing the virus was collected after 24 hr. In cases where the virus titer was low, the supernatant was concentrated approximately 25-fold using a filter unit with a nominal molecular weight limit of 100 kDa (UFC910024, AMICON). SFV particles encoding the respective Munc13 isoforms with EGFP or only EGFP as a control were added to chromaffin cell cultures, infected cells identified based on the EGFP fluorescence, and electrophysiological recordings performed 4–6 hr after addition of the virus.

## Whole-cell capacitance measurements

Whole cell patch-clamping was performed with Sylgard-coated 4–6 MΩ pipettes (Science Products) at a setup equipped with a Zeiss Axiovert 200 microscope (Zeiss) and a HEKA EPC-10 amplifier controlled by Patchmaster (HEKA). Capacitance measurements were performed according to the Lindau-Neher technique using the 'sine+dc' mode in the Lockin Extension of Patchmaster. The

frequency and peak-to-peak amplitude of the sine wave were 1042 Hz and 70 mV, respectively, and the holding potential was -70 mV. Recordings were sampled at 12.5 kHz and filtered at 2.9 kHz. Flash photolysis experiments were performed according to established protocols (*Voets, 2000*; *Walter et al., 2010*). The extracellular solution contained 147 mM NaCl, 10 mM HEPES, 11.1 mM D-glucose, 2.8 mM KCl, 2 mM $CaCl_2$, 1 mM $MgCl_2$ and 3 μM tetrodotoxin (pH 7.2, 300–310 mOsM). For flash experiments, the intracellular solution contained 109 mM L-glutamic acid, 35 mM HEPES, 5 mM nitrophenyl-EGTA (Synaptic Systems), 5.65 mM $CaCl_2$, 2 mM Mg-ATP, 0.3 mM Na-GTP, 0.205 mM fura-4F (Invitrogen), 0.3 mM furaptura (Invitrogen) and 1 mM ascorbic acid (titrated to pH 7.2 with CsOH, osmolarity 290–295 mOsM). The flash stimulus was applied approximately 80 s after the whole-cell configuration was established using a xenon lamp (Rapp OptoElectronics). Unless otherwise specified, only cells with pre-flash $[Ca^{2+}]$ in the range of 300–600 nM were used for analysis. In flash photolysis experiments requiring pre-flash $[Ca^{2+}]$ concentrations higher than 600 nM (*Figure 7* and *Figure 7—figure supplement 1*), pulses of light at wavelengths of 340 and 380 nm were applied at varying frequencies to release $Ca^{2+}$ from nitrophenyl-EGTA and the cell was kept at the target $[Ca^{2+}]$ for ~20 s before the flash stimulus was given. The pre-flash $[Ca^{2+}]$ was taken as the averaged measured $[Ca^{2+}]$ during the 20 s period. In depolarization experiments, the same extracellular solution was used except that tetrodotoxin was omitted. The intracellular solution contained 111 mM L-glutamic acid, 35.5 mM HEPES, 17 mM NaCl, 1 mM $MgCl_2$, 2 mM Mg-ATP, 0.3 mM Na-GTP (titrated to pH 7.2 with CsOH, osmolarity 290–295 mOsM), and fura-4F and furaptra at the same concentration used in flash experiments. In flash photolysis experiments, pool sizes (fast and slow bursts) and their time constants were obtained by fitting a sum of exponential functions to the capacitance traces (*Sorensen et al., 2003b*), using a custom macro (Three-Exponential-Fit-Macro-Igor) (see *Source code 1*) with the software IgorPro (WaveMetrics). The near-linear rate of release of the sustained component is measured as a linear component with the unit capacitance increase per second. The exocytotic delay was defined as the time point where the exponential fit meets the pre-flash capacitance.

## $Ca^{2+}$ measurements

In flash experiments, exocytosis was stimulated by a sudden elevation of intracellular $[Ca^{2+}]$ using UV flash stimuli given by a xenon flash lamp (Rapp OptoElectronics). $[Ca^{2+}]$ measurements were performed according to established protocols (*Voets, 2000*; *Walter et al., 2010*). The ratiometric $Ca^{2+}$ indicator dyes fura-4F and furaptra were alternately excited at 340 and 380 nm using a Polychrome V monochromator (TILL Photonics), and the emitted light was detected with a photomultiplier. The area of fluorescence measurement was limited to the diameter of the cell. The 340/380 nm fluorescence ratio was independently calibrated at the same dye concentrations with a range of intracellular solutions with known $[Ca^{2+}]$, buffered with $Ca^{2+}$ buffers 1,2-bis(o-aminophenoxy)ethane-N,N,N',N'-tetraacetic acid (BAPTA, Invitrogen) and diethylenetriaminepentaacetic acid (DPTA, Sigma-Aldrich). The $[Ca^{2+}]$ of the calibration solutions was calculated using $K_D$ of BAPTA = 0.222 μM and $K_D$ of DPTA = 80 μM.

## Amperometry

Amperometric recordings were performed using carbon fibers of 5 μm diameter (Thornel P-650/42; Cytec, NJ, USA), insulated using the polyethylene method (*Bruns, 2004*). Vesicle fusion was triggered by infusing the cells through the patch pipette with a solution containing 4.6 μM free $Ca^{2+}$. The fibers were clamped at 700 mV and currents were hardware filtered at 3 kHz using an EPC-7 patch clamp amplifier (HEKA). Currents were digitized at 25 kHz and filtered off-line using a Gaussian filter with a cut-off set at 1 kHz. Filtering, spike detection, and analysis were performed using a freely available, custom-written macro (*Mosharov and Sulzer, 2005*) running under IgorPro (Wavemetrics, Lake Oswego, OR). A spike detection threshold of 5 pA and a foot detection threshold of 2 pA were imposed. For each analyzed cell, the median of each parameter (duration, halftime, amplitude, charge, rise time, decay time, foot amplitude, foot duration, foot charge) was calculated from all spikes, and this value was used for averaging between cells (giving the mean of cell medians).

## High-pressure freezing of adrenal gland slices for EM analysis

Adrenal glands from E18 animals were embedded in 3% low gelling agarose (Sigma-Aldrich) and adrenal gland slices were prepared according to published protocols (*Moser and Neher, 1997*). Slices were allowed to recover in bicarbonate-buffered saline (125 mM NaCl, 26 mM NaHCO$_3$, 2.5 mM KCl, 1.25 mM NaH$_2$PO$_4$, 2 mM CaCl$_2$, 1 mM MgCl$_2$, 10 mM D-glucose and 0.2 mM (+)-tubocurarine) at 37°C for 15 min and were subsequently kept in the same solution at RT before cryofixation. Slices were rapidly frozen in external cryoprotectant (20% BSA in bicarbonate-buffered saline) using a HPM100 HPF device (Leica). After freezing, samples were stored in liquid nitrogen until further processing. Freeze substitution was performed as previously published (*Rostaing et al., 2006*). Briefly, samples were substituted in anhydrous acetone, fixed by 2% OsO$_4$ in acetone for 7 h at -90°C prior to a temperature ramp (5°C/h) to -20°C, an incubation for 16 hr at -20°C, and a final ramp (10°C/h) to 4°C. Samples were washed in acetone and infiltrated with EPON resin at room temperature (acetone/EPON 1:1 for 3 h, 90% EPON in acetone overnight, and pure EPON for 36 h). Finally specimen carriers containing infiltrated samples were incubated for 24 hr at 60°C to polymerize. Aluminum sample carriers were trimmed off the EPON block with a specimen trimming device (EM TRIM2, Leica) to expose the surface of the embedded tissue for ultramicrotomy.

## Sectioning, contrasting, and fiducial marker application for EM

An Ultracut UCT ultramicrotome (Leica) was used to cut 500 nm-thick sections until the first tissue appeared. Ultrathin (50 nm) and semithin (400 nm) sections were then collected onto Formvar-filmed, carbon-coated copper slot or mesh grids for 2D and 3D ultrastructural analysis, respectively. Vitrified samples were subjected to rigorous quality control (*Mobius et al., 2010*) and samples exhibiting indications of ice-crystal damage were excluded from the analysis. Ultrathin sections were post-stained with 1% uranyl acetate in ddH$_2$O for 30 min, washed several times in ddH$_2$O, stained with 0.3% lead citrate for 2 min, washed, and dried with filter paper. For 3D-tomographic analysis, 400 nm-thick sections were briefly incubated in a solution of Protein A conjugated to 10 nm gold particles (Cell Microscopy Center, Utrecht, The Netherlands) to introduce fiducial markers.

## 2D-EM analysis of chromaffin cells

Electron micrographs (2048 × 2048 pixels) were acquired with a sharp:eye CCD camera (Tröndle, TRS) at 5000 fold magnification using the multiple image acquisition and alignment feature of iTEM software (version 5.1, Olympus Soft Imaging Solutions). Assembled montages had dimensions of approximately 21 × 21 µm and typically contained 1–3 randomly chosen chromaffin cells. Only chromaffin cells with a clearly visible plasma membrane were included in the analysis. iTEM software was used to measure chromaffin cell plasma membrane circumference and the cytoplasmic area (calculated by subtraction of the nuclear area from the total cell area). Additional parameters including the number of LDCVs and the shortest distance of each LDCV to the plasma membrane were quantified using ImageJ software. In regions where the plasma membrane did not appear as a clear cut, the shortest distance from the vesicle membrane to the middle of the membrane projection was measured. For this reason and due to the fact that LDCVs (mean diameter of CTRL LDCVs ~160 nm; ~170 nm for docked CTRL LDCVs) might have their centers outside of the imaged ultrathin (50 nm) section, we did not quantify LDCV docking as defined by physical membrane-attachment in these 2D projection images, but rather calculated the number of membrane-proximal LDCVs (within 0–40 nm of the plasma membrane). All vesicles with electron-dense cargo were included in the analysis. Secretory vesicles of both genotypes exhibited heterogeneity in size (see *Figure 6—figure supplement 1*) and appearance (e.g. compactness of the dense-core), possibly reflecting distinct LDCV types (e.g. adrenaline vs. noradrenaline) or different levels of maturity present in immature (E18) chromaffin cells. Data are presented as LDCV density (number of LDCVs per µm$^2$ area cytoplasm), the number of LDCVs within 40 nm of the plasma membrane normalized to the cell perimeter, and the mean frequency distribution of LDCVs from the plasma membrane in 40 nm bins.

## 3D-EM analysis of LDCV docking in chromaffin cells

For high-resolution electron tomographic analysis of LDCV docking in adrenal chromaffin cells, we randomly selected regions between two neighboring cells that exhibited a high density of LDCVs in proximity of the plasma membrane in thick (400 nm) sections. Single-axis tilt series were acquired

from -60° to +60° with 1° increments and binned by the factor two at 10,000-fold magnification using an Orius SC1000 camera (Gatan) and the SerialEM software for automated tilt series acquisition (*Mastronarde, 2005*). Tomograms were reconstructed using the IMOD package (*Kremer et al., 1996*), and exported as z-stacks for analysis with ImageJ (National Institutes of Health). All analyses were performed blindly and manually. The smallest distance between the plasma membrane and the outer leaflet of each LDCV membrane was measured at its vesicular midline using the straight-line tool of ImageJ on individual virtual z-slices. Only vesicles observed in physical contact with the plasma membrane in tomographic volumes were considered 'docked'. In distribution analyses docked LDCVs were assigned to the 0–4 nm bin to account for the voxel dimensions of reconstructed tomograms (isotropic voxel size = 2.86 nm). The number of membrane-attached (0–4 nm, 'docked') LDCVs identified in a tomographically reconstructed volume was normalized to the number of membrane-proximal (0–40 nm of plasma membrane) LDCVs. The number of membrane-proximal (0–40 nm) LDCVs was expressed as a percentage of all LDCVs within 100 nm of the plasma membrane and compared across genotypes to test for potential differences in the ability of recruiting LDCVs close to the plasma membrane. The frequency distribution displays the number of docked LDCVs (0–4 nm, first bin) and subsequently the distances of LDCVs from the plasma membrane in 2 nm bins. Statistical analyses were performed using Student's $t$-test. The mean LDCV diameter was calculated from the area of the LDCV measured at its midline including the vesicular phospholipid bilayer in electron tomograms by using the elliptical selection tool in ImageJ. All LDCVs in the randomly chosen field of view that contained their midline within the tomographic volume were analyzed (*Figure 6—figure supplement 1*).

For illustrative purposes, figures depicting tomographic subvolumes represent an overlay of 3 consecutive slices produced by the slicer tool of the 3dmod software of the IMOD package to generate a ~8.6 nm thick subvolume.

## Calculation of the number of docked LDCVs per chromaffin cell

The size of the pool of docked LDCVs per cell can be calculated from the number of LDCVs per $\mu m^2$ PM area ($n_a$) (*Parsons et al., 1995*; *Plattner et al., 1997*). We measured the number of membrane-proximal LDCVs (0–40 nm of PM) per $\mu m$ PM length ($n_l$) in ultrathin sections of 0.05 $\mu m$ thickness. Our 3D-ET approach permitted us to accurately measure LDCV diameters ($d_v$ in $\mu m$) in chromaffin cells (*Figure 6—figure supplement 1*). The number of LDCVs per $\mu m^2$ PM area could then be calculated as $n_a = n_l/(d_v + 0.05)$ (*Parsons et al., 1995*; *Plattner et al., 1997*). The average cell surface area ($a_c$ in $\mu m^2$) per genotype was estimated based on the average cell capacitance measured in cultured cells assuming 1 $\mu F/cm^2$. We chose this method, rather than using the cell circumference measured in ultrathin sections, because the chromaffin cells in adrenal slices are not round, but have rather complex shapes (*Figure 6—figure supplement 1*). Using either measurement, the *Unc13a*[KO]*Unc13b*[KO] cells were slightly larger, by 5% based on capacitance, and by 15% based on cell circumference measurements.

*Unc13a*[Het]*Unc13b*[Het] control cells:
$n_l$: 0.984 ± 0.075 vesicles per $\mu m$ length
$d_v$: 0.1627 ± 0.005 $\mu m$ vesicle diameter
$n_a$: 4.627 vesicles per $\mu m^2$ PM area
$a_c$: 421.23 $\mu m^2$ cell membrane area

The estimated number of LDCVs within 40 nm of the PM in *Unc13a*[Het]*Unc13b*[Het] chromaffin cells is therefore ~1949. Our 3D analysis of LDCV docking revealed that only 33.97% of LDCVs within 40 nm of the PM are physically attached to the PM, therefore we estimated the pool of docked vesicles to contain ~662 LDCVs in our acute adrenal slice preparation.

*Unc13a*[KO]*Unc13b*[KO] cells:
$n_l$: 1.150 ± 0.083 vesicles per $\mu m$ length
$d_v$: 0.1525 ± 0.003 $\mu m$ vesicle diameter
$n_a$: 5.679 vesicles per $\mu m^2$ PM area
$a_c$: 442.26 $\mu m^2$ cell membrane area

The estimated number of LDCVs within 40 nm of the PM in *Unc13a*[KO]*Unc13b*[KO] chromaffin cells is therefore ~2511. Our 3D analysis of LDCV docking revealed that only 34.43% of LDCVs within 40 nm of the PM are physically attached to the PM, therefore we estimated the pool of docked vesicles to contain ~865 LDCVs in our acute adrenal slice preparation.

## Modeling the Ca²⁺-dependence of vesicle priming

We simulated capacitance traces in Ca²⁺ uncaging experiments with an exocytosis model that was adapted from a previous study of ours (*Walter et al., 2013*) to explicitly describe the Ca²⁺-dependent vesicle priming reaction. Parameters (*Table 1*) were either taken from the literature, directly from experiments (*Figure 7A*), or determined by fitting the model to experimental capacitance traces.

We assume that Munc13 proteins act on the Ca²⁺-dependent priming reaction, which ensures the refilling of vesicles from a large depot pool (*Voets, 2000*). We wanted to explicitly describe this reaction in terms of its thermodynamic steady state binding properties (i.e. its dissociation constant $K_D$ and cooperativity n), and in terms of its Ca²⁺ binding kinetics. Let the chemical equation for Ca²⁺ binding to the priming sensor (PS) be:

$$n * Ca^{2+} + PS \underset{k_{off}}{\overset{k_{on}}{\rightleftharpoons}} (Ca_nPS)$$

Then its overall dissociation constant $K_D$ is defined as:

$$K_D = \frac{[Ca^{2+}]^n [PS]}{[Ca_nPS]}$$

In order to obtain the values of the $K_D$ and n experimentally, we pre-equilibrated chromaffin cells at different pre-flash [Ca²⁺] prior to uncaging. The secretion responses were averaged in bins and normalized to their respective values 3 s after the flash. Fitting the fraction of release 30 ms after the uncaging stimulus as a function of pre flash [Ca²⁺] with the following Hill equation (non-linear curve fitting routine of Origin Pro 8 G, OriginLab Corporation) allowed us to estimate $K_D$ and n:

$$Fraction([Ca^{2+}]) = \frac{[Ca^{2+}]^n}{K_D + [Ca^{2+}]^n} Fmax$$

Where $Fraction([Ca^{2+}])$ is the relative release at 30 ms, and [Ca²⁺] is the pre-flash Ca²⁺ concentration. $Fmax$, n and $K_D$ are free parameters. The best fit is shown as solid line in *Figure 7* and the values of n and $K_D$ can be found in *Table 1*.

We assume that priming is only increased when the proper number (n) of Ca²⁺ ions are bound to the PS. Therefore, the rate of priming is proportional to the fraction ($f$) of PS that has bound the correct number of Ca²⁺ ions divided by the total amount of PS:

$$f(Ca^{2+}) = \frac{[Ca_nPS]}{[PS_{tot}]}$$

$$k_1(Ca^{2+}) = f(Ca^{2+}) \, k_{1Max}$$

Where $k_{1Max}$ is the asymptotic value of the priming rate for $[Ca^{2+}] \gg K_D$. Since the total amount of PS is the sum of Ca²⁺-free and Ca²⁺-bound PS:

$$f(Ca^{2+}) = \frac{[Ca_nPS]}{[Ca_nPS] + [PS]}$$

At steady state, the following relationships hold:

$$f(Ca^{2+})_{SteadyState} = \frac{[Ca^{2+}]^n}{K_D + [Ca^{2+}]^n}$$

$$k_{1SteadyState}(Ca^{2+}) = \frac{[Ca^{2+}]^n}{K_D + [Ca^{2+}]^n} k_{1Max}$$

In order to describe temporal changes in this fraction at non-equilibrium conditions, the Ca²⁺ concentrations ($[Ca^{2+}]$) were interpolated from the experimental values and the concentration of $[Ca_nPS]$ was calculated at all time points by numerical integration using the 'ode15s' function in Matlab (version R2013a, Mathworks) of the kinetic equation:

$$\frac{d[Ca_nPS]}{dt} = k_{on}[PS][Ca^{2+}]^n - k_{off}[Ca_nPS]$$

We assume that the amount of PS is not limiting (i.e. that each vesicle contains a PS). Then, by investing the relationships

$$[V_{tot}] = [Ca_nPS] + [PS]$$

and

$$K_D = \frac{k_{off}}{k_{on}}$$

we obtain

$$\frac{d[Ca_nPS]}{dt} = k_{on}([V_{tot}] - [Ca_nPS])[Ca^{2+}]^n - K_D k_{on}[Ca_nPS]$$

Such that the fraction of activated PS can be calculated at time t:

$$f(Ca^{2+}, t) = \frac{[Ca_nPS](t)}{[V_{tot}]}$$

This allows us to calculate $k_1$ at all times t.

$$k_1(Ca^{2+}, t) = f(Ca^{2+}, t) k_{1Max}$$

Our model consists of a sequence of mandatory steps for vesicle maturation and fusion (*Walter et al., 2013*). Vesicles from the depot enter a non-releasable state (NRP, *Figure 7E*) from which they cannot fuse directly. Instead, these vesicles first need to mature into the readily releasable pool (RRP), a transition that is governed by a $Ca^{2+}$-dependent rate constant ($k_2[Ca^{2+}]$). This $Ca^{2+}$-dependence is modeled by a $Ca^{2+}$-dependent catalyst as described in *Walter et al., 2013*:

$$\begin{aligned} k_2(Ca^{2+}) &= k_{20} + g(Ca^{2+})k_{2cat} \\ k_{-2}(Ca^{2+}) &= k_{-20} + g(Ca^{2+})k_{-2cat} \\ k_{-2cat} &= \frac{k_{-20}}{k_{20}}k_{2cat} \end{aligned}$$

As in our previous study, we assume that binding of one $Ca^{2+}$ ion activates the catalyst and that the catalyst is in equilibrium with $Ca^{2+}$, which allows us to calculate $g(Ca^{2+})$:

$$g(Ca^{2+}) = \frac{[Ca^{2+}]}{K_{D,cat} + [Ca^{2+}]}$$

**Kinetic equations of the exocytosis model**

$$\frac{d[Depot]}{dt} = -k_1(Ca^{2+})[Depot] + k_{-1}[NRP]$$

$$\frac{d[NRP]}{dt} = k_1(Ca^{2+})[Depot] - (k_{-1} + k_2(Ca^{2+}))[NRP] + k_{-2}(Ca^{2+})[RRP]$$

$$\frac{d[RRP]}{dt} = k_2(Ca^{2+})[NRP] - (k_{-2}(Ca^{2+}) + 3k_3[Ca^{2+}])[RRP] + k_{-3}[RRPCa]$$

$$\frac{d[RRPCa]}{dt} = 3k_3[Ca^{2+}][RRP] - (k_{-3} + 2k_3[Ca^{2+}])[RRPCa] + 2k_{-3}[RRPCa_2]$$

$$\frac{d[RRPCa_2]}{dt} = 2k_3[Ca^{2+}][RRPCa] - (2k_{-3} + k_3[Ca^{2+}])[RRPCa_2] + 3k_{-3}[RRPCa_3]$$

$$\frac{d[RRPCa_3]}{dt} = k3[Ca^{2+}][RRPCa_2] - (3K_{-3} + k_4)[RRPCa_3]$$

$$\frac{d[F]}{dt} = k_4[RRPCa_3]$$

### Modeling of exocytosis upon Ca$^{2+}$ uncaging

To find the steady state occupation of the system $k_1(Ca^{2+})$, $k_2(Ca^{2+})$ and $k_{-2}(Ca^{2+})$ were calculated for the experimental pre-flash Ca$^{2+}$ values, the first five kinetic equations were taken and set to zero, and mass conservation of vesicles was obeyed:

$$0 = [Depot]_0 + [NRP]_0 + [RRP]_0 + [RRPCa]_0 + [RRPCa_2]_0 + [RRPCa_3]_0 - [V_{tot}]$$

This system of 6 equations was solved using the function 'fsolve' of Matlab (version R2013a, Mathworks). The steady state values were used as starting values for the numerical integration of the kinetic equations using the 'ode15s' function of Matlab (version R2013a, Mathworks).

For parameter optimization, model simulations were compared to experimental capacitance data. The parameters of the model were varied and the goodness of fit was determined in a cost function, which was the sum of squared deviations between data and model prediction. In order to avoid bias towards data with larger secretion, the cost function used relative deviations: deviations were normalized to the maximal value of either the experimental capacitance trace or the simulated one, whichever was smaller (led to a larger cost):

$$cost = \frac{weight(i)}{\max(y_{Experiment})} \sum_i \left( y_{Experiment}(i) - y_{Model}(i) \right)^2 for \max(y_{Experiment}) < \max(y_{Model})$$

$$cost = \frac{weight(i)}{max(y_{Model})} \sum_i \left( y_{Experiment}(i) - y_{Model}(i) \right)^2 for \max(y_{Experiment}) \geq max(y_{Model})$$

Three principal kinetic components have been described in capacitance traces: a fast component with a time constant of tens of milliseconds, a slower component with a time constant of hundreds of milliseconds and a sustained component with a time constant of several seconds (*Voets, 2000*). To account for the fact that more data points exist for slower components (due to the constant sampling rate) which would dominate the fit, deviations at shorter times after the uncaging flash were given larger weight: the weight was 100 for all datapoints upto 80 ms after the uncaging stimulus; the weight was 10 for all datapoints occurring later than 80 ms, but earlier than 1.2 s after the stimulus; the weight was 1 for all datapoints thereafter. To ensure consistency, all conditions depicted in *Figure 7* were fitted simultaneously and the cost values from all data were summed. Only the parameters labeled with "best fit" in *Table 1* were allowed to vary. Because the lowest bin of the capacitance traces of the DKO overexpressing Munc13-1 and ubMunc13-2 in *Figure 7B and C* contained relatively few cells, the total number of vesicles was also a free parameter under these conditions (best fit values V$_{tot}$(Munc13-1 low Ca$^{2+}$) = 2120 fF, V$_{tot}$(Munc13-1 low Ca$^{2+}$) = 3340 fF). All

parameters were optimized by minimizing the cost values using a Nelder-Mead Simplex algorithm implemented in the Matlab function 'fminsearch' (version R2013a, Mathworks).

## Statistics

Statistical analyses were performed using two-tailed Student's *t*-test, ANOVA with post-hoc Tukey's test, or Mann-Whitney U-test as specified in the figure legends.

## Acknowledgements

This work was supported by the Max Planck Society. KNM was supported by the German Academic Exchange Service and the Göttingen Graduate School for Neurosciences, Biophysics and Molecular Biosciences. AW was supported by the Emmy Noether Program of the Deutsche Forschungsgemeinschaft. The work was supported by The Danish Council for Independent Research in Medical Sciences (JBS). We thank JS Rhee for advice, A Zeuch, A Günther, I Beulshausen, S Bolte, F Benseler, I Thanhäuser, D Schwerdtfeger, C Harenberg, and M Schlieper for excellent technical support, and the MPIEM animal facility for mouse husbandry.

## Additional information

### Funding

| Funder | Author |
| --- | --- |
| Max-Planck-Gesellschaft | Kwun Nok M Man<br>Cordelia Imig<br>Benjamin H Cooper<br>Nils Brose<br>Sonja M Wojcik |
| Deutsche Forschungsgemeinschaft | Alexander Matthias Walter |

The funders had no role in study design, data collection and interpretation, or the decision to submit the work for publication.

### Author contributions

KNMM, CI, Conception and design, Acquisition of data, Analysis and interpretation of data, Drafting or revising the article; AMW, Analysis and interpretation of data, Drafting or revising the article; PSP, Acquisition of data, Analysis and interpretation of data; DRS, JR, Analysis and interpretation of data, Contributed unpublished essential data or reagents; JBS, Conception and design, Analysis and interpretation of data, Drafting or revising the article; BHC, Conception and design, Acquisition of data; NB, Conception and design, Drafting or revising the article; SMW, Conception and design, Wrote the paper, Drafting or revising the article, Analysis and interpretation of data

### Ethics

Animal experimentation: All experiments were performed in compliance with the regulations of the local Animal Care and Use Committee of Lower Saxony, Oldenburg, Germany.

## Additional files

### Supplementary files

• Source code 1. Custom macro: Three-Exponential-Fit-Macro-Igor. Macro for kinetic analysis of chromaffin cell capacitance traces developed with Igor Pro v6.32A (WaveMetrics) and Patcher's Power Tools v2.19 (http://www3.mpibpc.mpg.de/groups/neher/index.php?page=software). Capacitance changes were triggered with calcium uncaging and the traces recorded with PatchMaster v2.20 (HEKA).

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
