## [Decision Letter]

Thank you for submitting your work entitled "Identification of a Munc13-sensitive step in chromaffin cell large dense-core vesicle exocytosis" for peer review at *eLife*. Your submission has been favorably evaluated by Gary Westbrook (Senior editor) and three reviewers, one of whom is a member of our Board of Reviewing Editors. One of the three reviewers has agreed to reveal his identity: Josep Rizo (Reviewer 2).

The reviewers have discussed the reviews with one another and the Reviewing editor has drafted this decision to help you prepare a revised submission. This is a thorough study (using capacitance, amperometry, and EM on chromaffin cells from various mouse Munc13 KO lines) of the role of Munc13 in large dense core vesicle (LDCV) exocytosis in chromaffin cells. Previous studies had shown that Munc13s and CAPSs, proteins that like Munc13 contain a MUN domain, play important functions in synaptic vesicle exocytosis in mammals, and CAPSs had also been shown to mediate LDCV in chromaffin cells. However, studies in invertebrates showed that synaptic vesicle exocytosis requires the Munc13 homologue Unc13 while LDCV exocytosis requires the CAPS homologue unc31. The key question that arises is whether Munc13s are involved in LDCV exocytosis in chromaffin cells. The authors show that Munc13-2 plays a key role in priming, but not docking of LDCVs. This is at variance of the role of Munc13 in synaptic vesicle exocytosis where it also appears to also play a role in docking. The most striking result is that LDCV docking in chromaffin cells does not require Munc13 in marked contrast to synaptic vesicles (compare Figure 6 with Figure 2 of Imig et al., Neuron, 2014).

While the authors present the data as indicating major differences between SV and LDCV exocytosis (Munc13s as critical for LDCV priming but not docking), this conclusion is highly qualified by the likelihood that the pool of primed LDCVs is very small relative to the number of docked LDCVs in contrast to pools of primed SVs. An additional complication is that various protocols involve a calcium-dependent activation step so that an RRP in resting cells is not really being measured. The location of Munc13s relative to sites of exocytosis is not indicated. Moreover, there are some concerns about the discussion and interpretation of the results.

1) The authors propose a model of LDCV docking and priming: "(i) LDCV docking mediated by Munc18-1/Stx-1, this configuration would be the starting point for Munc13-mediated SNARE complex assembly, i.e. priming (Ma et al., 2011; Ma et al., 2013), and in parallel, (ii) LDCV docking mediated by Syt-1/Stx-1/SNAP-25, assuming that in chromaffin cells Stx-1/SNAP-25 dimers form spontaneously, as they do in vitro (Ma et al., 2013)". However, while Munc18 interactions with membranes have been reported (e.g., Xu et al., PLOS One 6, e22012, 2011) such interactions appear to be rather non-specific. Another possibility is binding of Munc18 to Rab3, i.e., docking vial Rab3/Munc18-1/Stx-1. It is less likely that Syt1 were to facilitate docking via interaction with the syntaxin-SNAP-25 (binary) complex, since it would be inconsistent with the results by Ma et al., 2013 that suggests that NSF/SNAP disassembles the binary complex, allowing Munc18 to strongly bind to syntaxin and inducing the closed conformation of syntaxin. While these other models and experimental results do not necessarily rule out the models proposed by the authors, there appears to be some uncertainty in interpreting the observed results in terms of molecular interactions. A more complete discussion would be desirable, and perhaps offering some other explanations as well (e.g., the existence of a new tethering factor or tethering/docking mechanism for LDCV vesicles).

2) The interpretation of the results in Figure 7 in terms of SNARE complex assembly is only one possible explanation. Another possible explanation would be the existence of two different pools of vesicles that fuse with different rates and that are affected differently by overexpression of Munc13-1/2.

3) Figure 5: The authors compared viral overexpression of Munc13 isoforms in rescue studies with KO cells. They show that ubMunc13-2 exhibits more activity than Munc13-1. However, the relative amounts of expressed protein are not shown making it difficult to assess actual intrinsic activity. Similarly the apparent lack of activity of Baiap3 or reduced activity of Munc13-4 could be due to inefficient expression of these proteins. Was the EGFP fluorescence of these expressed constructs used to normalize the values recorded or otherwise used to assess expression levels?

4) Figure 6: The studies shown in this figure are featured in the Abstract to indicate that "molecular requirements of morphological LDCV and SV docking are distinct." This conclusion needs more discussion given the authors' data that a loss of RRP vesicles may not be detected within the 20-fold larger pool of docked LDCVs. Failure to detect a difference in KO cells leaves open the possibility that SV and LDCV RRP pools are mechanistically similar but that non-RRP docked LDCVs may be docked by a separate mechanism whereas most docked SVs are in the RRP.

5) The protocols for measuring the RRP and morphologically docked LDCVs differ in important ways. Protocols for the RRP seem to involve a calcium-dependent pre-priming step that is absent in the morphological studies. Have the authors considered the possibility that the EM studies might show more docked LDCVs if there was a pre-priming step and this may be missing in the KO cells?

6) In a new model (Walter et al., 2013), the authors conceptualize slow burst and fast burst as representing sequential NRP and RRP pools. They then reach the conclusion that Munc13 alters k_1_ in "priming 1" step. However, it seems that the experimental fitting started with this assumption. Paradoxically k_1_ and k_2_ in the model changed in opposite directions, an outcome that is difficult to understand. It seems that other models would also fit the data but these are not discussed. Priming 1 and priming 2 might correspond to parallel pathways of priming and fusion. Several studies suggest two classes of chromaffin granules based on size or synaptotagmin composition. Another very possible model would be that k_1_ corresponds to the translocation of Munc13 to the membrane and is enhanced by calcium-dependent pre-priming. Thus, it is difficult to be convinced that the formation of the NRP is a precursor to the RRP as part of a two-step priming mechanism or that Munc13 affects k_1_ but not k_2_. Alternative hypotheses for the NRP and RRP need discussion.

7) Betz et al. 1998 showed that Munc13-1 was cytosolic. Multiple modes for calcium regulation of Munc13-1/2 have been reported by the Brose lab and others. Localization studies especially with the more abundant ubMunc13-2 in resting or calcium-primed cells would provide much needed clarification to the proposed model. Is Munc13 at sites of exocytosis in resting cells? Does it undergo translocation to sites of exocytosis in calcium-dependent pre-priming? This knowledge may affect the model, which is currently based on vesicle pools rather than the overall preparedness of the exocytic machinery.

8) There seems to be over-reliance on a particular model to interpret data. "Using a mathematical model that interprets the kinetic phases of LDCV release as phases of SNARE complex assembly" as stated in the Abstract does not seem to describe this aspect of the work accurately. Although assembly of an N-terminal SNARE complex is likely important in priming, there is yet no evidence that Munc13s uniquely catalyzes this reaction. The current studies do not address that issue.

9) Please rephrase the following passage: "Raising the possibility that the functions of different Munc13 isoforms in priming LDCVs and SVs are conserved, even though their role in docking is not". This needs to be qualified by the detection issue. There may be two modes of docking as the authors discuss but Munc13s might function in one of these modes similar to that for SVs.

---

## [Author Response]

1) The authors propose a model of LDCV docking and priming: "(i) LDCV docking mediated by Munc18-1/Stx-1, this configuration would be the starting point for Munc13-mediated SNARE complex assembly, i.e. priming (Ma et al., 2011; Ma et al., 2013), and in parallel, (ii) LDCV docking mediated by Syt-1/Stx-1/SNAP-25, assuming that in chromaffin cells Stx-1/SNAP-25 dimers form spontaneously, as they do in vitro (Ma et al., 2013)". However, while Munc18 interactions with membranes have been reported (e.g., Xu et al., PLOS One 6, e22012, 2011) such interactions appear to be rather non-specific. Another possibility is binding of Munc18 to Rab3, i.e., docking vial Rab3/Munc18-1/Stx-1. It is less likely that Syt1 were to facilitate docking via interaction with the syntaxin-SNAP-25 (binary) complex, since it would be inconsistent with the results by Ma et al., 2013 that suggests that NSF/SNAP disassembles the binary complex, allowing Munc18 to strongly bind to syntaxin and inducing the closed conformation of syntaxin. While these other models and experimental results do not necessarily rule out the models proposed by the authors, there appears to be some uncertainty in interpreting the observed results in terms of molecular interactions. A more complete discussion would be desirable, and perhaps offering some other explanations as well (e.g., the existence of a new tethering factor or tethering/docking mechanism for LDCV vesicles).

We thank the reviewers for this comment, we have modified the Discussion section to better reflect the uncertainty with respect to the molecular interactions involved in LDCV docking, particularly where Syt1, Stx-1 and SNAP-25 are involved, and point out the possibility that additional factors may be involved in docking.

2) The interpretation of the results in Figure 7 in terms of SNARE complex assembly is only one possible explanation. Another possible explanation would be the existence of two different pools of vesicles that fuse with different rates and that are affected differently by overexpression of Munc13-1/2.

The question of whether the two downstream pools (RRP and NRP/SRP) can both fuse is also raised in point 6 – see response to point 6 for an answer. Please note that if those two pools could both fuse (RRP, SRP) and if they were primed via parallel routes, then the major effect of Munc13-1/ubMunc13-2 would be the same on both pools (except for a small difference, encoded currently in the changed parameter k_2_). Thus, we would need a separate set of almost identical parameters for each pool. In our model, we offer an elegant explanation: that the priming step where Munc13-1/2 has its major effect (priming 1) is upstream of both pools (SRP/NRP and RRP). This is supported by other, classical, experiments as well (see answer to point 6 below).

3) Figure 5: The authors compared viral overexpression of Munc13 isoforms in rescue studies with KO cells. They show that ubMunc13-2 exhibits more activity than Munc13-1. However, the relative amounts of expressed protein are not shown making it difficult to assess actual intrinsic activity. Similarly the apparent lack of activity of Baiap3 or reduced activity of Munc13-4 could be due to inefficient expression of these proteins. Was the EGFP fluorescence of these expressed constructs used to normalize the values recorded or otherwise used to assess expression levels?

We thank the reviewers for this comment. To confirm that the apparent intrinsic differences in the ability of Munc13-1 and ubMunc13-2 to promote LDCV secretion are not merely due to higher expression levels of ubMunc13-2, we have added a supplemental figure (Figure 5—figure supplement 1), where only cells of matched low to medium fluorescence intensities were analyzed. Based on this direct comparison, cells expressing ubMunc13-2-EGFP still show significantly larger burst sizes (*p* < 0.01) and rates of sustained release (*p* < 0.001) than cells expressing Munc13-1-EGFP.

However, we chose not to normalize all capacitance data based on fluorescence levels, because the fluorescence intensity does not appear to scale with the rate of secretion in a linear fashion.

Furthermore, while Munc13-1 and ubMunc13-2 were expressed as EGFP fusion constructs whose function is identical to that of WT constructs as tested in neurons (Rosenmund et al., Neuron, 2002, 33, 411-424), we chose to express Baiap3 and Munc13-4 as internal ribosome entry site (IRES) constructs, because for these two isoforms we had no way of confirming that an EGFP-tag would not interfere with protein function. We did however confirm that our Semliki Forest Virus preparations in fact express Baiap3 and Munc13-4. We have added the corresponding Western blots, which were done with infected neurons, because our chromaffin cell cultures do not provide enough material for Western blotting, as a supplemental figure (Figure 5—figure supplement 2).

4) Figure 6: The studies shown in this figure are featured in the Abstract to indicate that "molecular requirements of morphological LDCV and SV docking are distinct." This conclusion needs more discussion given the authors' data that a loss of RRP vesicles may not be detected within the 20-fold larger pool of docked LDCVs. Failure to detect a difference in KO cells leaves open the possibility that SV and LDCV RRP pools are mechanistically similar but that non-RRP docked LDCVs may be docked by a separate mechanism whereas most docked SVs are in the RRP.

We agree with the reviewers. However, even if the RRP is physically rather than just functionally “lost” in the absence of Munc13-1 and Munc13-2, the presence of the docked non-RRP pool in itself implies that the docking mechanisms of SVs (which do not dock in the absence of Munc13s) and LDCVs (which do) must be distinct. Moreover, we in fact do think that the formation of the RRP is mechanistically similar for SV and LDCVs, whereas the non-RRP docked LDCVs are docked via a separate mechanism. We now discuss possible mechanisms of RRP and non-RRP LDCV docking in more detail.

5) The protocols for measuring the RRP and morphologically docked LDCVs differ in important ways. Protocols for the RRP seem to involve a calcium-dependent pre-priming step that is absent in the morphological studies. Have the authors considered the possibility that the EM studies might show more docked LDCVs if there was a pre-priming step and this may be missing in the KO cells?

This is of course a valid point, and we have considered this possibility. However, the docked pool detected under resting conditions is already much larger than the physiological RRP detected in stimulated cells. Since one of the conclusions of our tomographic 3D analysis is that the RRP, i.e. the pool where Munc13s play a role, cannot be distinguished from other docked LDCVs, we feel that even if we detect more docked LDCVs in stimulated cells, such a finding would add only a limited amount of information to the current study. Moreover, we expect that establishing a reliable stimulation protocol in combination with high-pressure-freezing, plus analyzing a sufficient number of samples with tomography would take us at least 5-6 months, and thus by far exceed the time frame of the revision period. In view of these arguments, we respectfully ask that the requirement to do such experiments be waived.

6) In a new model (Walter et al., 2013), the authors conceptualize slow burst and fast burst as representing sequential NRP and RRP pools. They then reach the conclusion that Munc13 alters k_1_ in "priming 1" step. However, it seems that the experimental fitting started with this assumption. Paradoxically k_1_ and k_2_ in the model changed in opposite directions, an outcome that is difficult to understand. It seems that other models would also fit the data but these are not discussed. Priming 1 and priming 2 might correspond to parallel pathways of priming and fusion. Several studies suggest two classes of chromaffin granules based on size or synaptotagmin composition.

There are a lot of separate issues in this question, which are concerned with why the model looks as it does. One issue is whether the pools we call NRP and RRP might fuse independently, as a slowly and a readily releasable pool, respectively, as in previous models (Voets, 2000). We (or better, some of us) have published data to support the idea that the slow pool fuses via the fast pool (Walter et al., 2013). We sense that the reviewers likely disagree, however for the present paper this issue does not matter much because we conclude that ubMunc13-2 and Munc13-1 exert their main effect (on k_1_) upstream of both pools. Therefore, whether or not the NRP can fuse is unlikely to change the conclusion of our modeling very much – although, of course the parameters will be changed. We have modified the Discussion to indicate that the data in the present manuscript are consistent with both ideas.

Another issue is whether the two pools (NRP and RRP in our notation) are filled by sequential or parallel pathways, i.e. whether priming 1 and 2 should be parallel or sequential reactions. Here, we rely on classical data from Voets et al. 1999 (Figure 5), showing that the RRP refills at the expense of the SRP (NRP in our model). This finding was the basis for the sequential arrangement of SRP and RRP in 1999. This was further substantiated by the fact that the Ca^2+^-dependent priming is similar for SRP and RRP, and therefore it is reasonable to assume a common upstream step feeding both pools (Voets, 2000). If it can be shown that the two types of vesicles are molecularly different – and if molecular differences between vesicles can be related directly to those two phases in capacitance traces, and if some other explanation for the effects in the Voets papers can be found – this would have to be changed. However, for now we keep the same assumption as in previous papers, since our present paper does not address this question.

The reason that our model starts with the assumption of k_1_ being changed by Munc13-1/ubMunc13-2 is because both the fast and slow burst (originating from the RRP and NRP/SRP, respectively) are changed almost proportionally upon Munc13-2 knockout/overexpression – and given that the same goes for the sustained component – this immediately identifies k_1_ as the critical parameter, since only a changed k_1_ would have these effects. So, even though we have to make the assumption up front (which is a basic premise of modeling), this is a very clear choice in this case.

We do not agree that there is anything paradoxical in our finding that Munc13 proteins exert opposite effects on k_1_ and k_2_. Please note that the major effect is on k_1_, and the effect on k_2_ – although it is consistent – is comparatively minor. In fact, we think it is an elegant feature of our modeling that by doing a simultaneous fit of the model to all conditions, we arrive at one set of consistent parameters that can detect this novel effect. This would be impossible to detect (but easy to gloss over) if we had fitted each condition independently, as some researchers did in the past. We put forward a reasonable explanation: “that the interaction of Munc13s with the SNARE fusion machinery may compete with that of another priming protein, which mainly catalyzes priming step 2.” Thus, it seems to be a general feature of most proteins involved in exocytosis that they exert both positive and negative effects. This has been demonstrated for Munc18-1, complexin, synaptotagmin and even for the SNAREs.

[…] Another very possible model would be that k_1_ corresponds to the translocation of Munc13 to the membrane and is enhanced by calcium-dependent pre-priming. Thus, it is difficult to be convinced that the formation of the NRP is a precursor to the RRP as part of a two-step priming mechanism or that Munc13 affects k_1_ but not k_2_. Alternative hypotheses for the NRP and RRP need discussion.

We agree with the reviewers, it is entirely possible – maybe even likely – that the Ca^2+^-binding to Munc13-1 and ubMunc13-2 (reactions shown to the left of k_1_ in Figure 7) are linked to the translocation of Munc13-1/ubMunc13-2 to the membrane, as a prerequisite for the action of Munc13s on vesicle priming (k_1_). We now mention this explicitly in the paper and thank the reviewers for pointing this out. Please note that this is not another model – it is the same model as the one we suggest, only with an additional interpretation.

The question of model discrimination (which model describes the data better? – which model requires fewer free parameters to fit the data?) is extremely complicated, since different models are not necessarily subsets of each other. To distinguish between a range of models is therefore a major challenge, and it is not even clear that it would be possible, or that enough data exist to perform this distinction. Thus, as is common practice in the field, we made an informed guess about the correct model (and in the process, we have explored other options, but going through all of them would make the paper impossible to read). The elegance (we feel) of the model is that only a very small set of different parameters are necessary to describe the functional difference between ubMunc13-2 and Munc13-1. Please note that the major effect on priming is covered by only two different parameters between ubMunc13-2 and Munc13-1 (the energy barrier for Ca^2+^ binding, and the different effects on k_1_), and that both ubMunc13-2 and Munc13-1 perform similar functions on the fusion machinery (increasing k_1_, most likely by stimulating SNARE-complex assembly). This is nicely consistent with the expectation that two so similar proteins should have similar effects. Yet, the model comes up with a useful prediction about why the kinetic outcomes can be different. This prediction can be tested. We think our model goes further towards a consistent picture of the involvement of Munc13 isoform in secretion than previous attempts.

7) Betz et al. 1998 showed that Munc13-1 was cytosolic. Multiple modes for calcium regulation of Munc13-1/2 have been reported by the Brose lab and others. Localization studies especially with the more abundant ubMunc13-2 in resting or calcium-primed cells would provide much needed clarification to the proposed model. Is Munc13 at sites of exocytosis in resting cells? Does it undergo translocation to sites of exocytosis in calcium-dependent pre-priming? This knowledge may affect the model, which is currently based on vesicle pools rather than the overall preparedness of the exocytic machinery.

Please also see response to point 6. We agree that it is indeed rather plausible that Ca^2+^-binding to Munc13-1 and ubMunc13-2, i.e. the reactions shown to the left of k_1_ in Figure 7, are linked to the translocation of Munc13-1/ubMunc13-2 to the membrane, as a prerequisite for the action of Munc13s on vesicle priming (k_1_). We have modified the text to add this interpretation and to explicitly state that the differences in delay might reflect differences in the membrane translocation step.

With respect to the question of whether endogenous Munc13 localizes to release sites in resting cells, this would indeed be very interesting, and we have tried multiple times to determine the localization of individual endogenous Munc13-isoforms in chromaffin cells. We initially tested isoform- specific antibodies in WT chromaffin cells, using the respective KO lines as controls. However, all antibodies to Munc13-1, ubMunc13-2 and Baiap3 that we tested resulted in non-specific, punctate labeling in both WT and KO cells, which made it impossible to detect any “real” signal. We now tried to circumvent this problem by using chromaffin cells from Munc13-2-EYFP knockin (KI) mice, which express Munc13-2-EYFP from the endogenous locus. Since the EYFP-signal was too weak, we used an antibody that detects the GFP-derived tag. In this case, we used WT cells as a negative control. Somewhat to our surprise, we again found strong, punctate labeling with the GFP-antibody in both Munc13-2-EYFP-KI and WT cells.

We currently do not know the origin of this non-specific punctate staining, but believe that it could be due to non-specific antibody trapping in chromogranin-negative vesicles during the various staining procedures employed. Unfortunately, we are therefore currently unable to reliably determine the localization of the endogenous Munc13 isoforms in mouse chromaffin cells. In view of this fundamental problem, we ask respectfully that the requirement for further experiments in this regard be waived.

8) There seems to be over-reliance on a particular model to interpret data. "Using a mathematical model that interprets the kinetic phases of LDCV release as phases of SNARE complex assembly" as stated in the Abstract does not seem to describe this aspect of the work accurately. Although assembly of an N-terminal SNARE complex is likely important in priming, there is yet no evidence that Munc13s uniquely catalyzes this reaction. The current studies do not address that issue.

We agree with the reviewers that we should phrase this more carefully, as our model does not directly model the SNAREs. We have now changed the Abstract to remove the words “SNARE complex assembly”. However, the point that we do want to make is that the very same priming step that is mainly affected by Munc13-1/ubMunc13-2 (k_1_) is also affected when mutating the N-terminal part of the SNAREs (as we have shown previously). This is consistent with biochemical experiments showing that the Mun-domain of Munc13 stimulates SNARE-complex assembly (for instance the very recent paper by Yang et al., 2015) and, therefore, allows us to make the suggestion that Munc13-1/ubMunc13-2 stimulates N-terminal SNARE-complex assembly. We have modified the corresponding section in the Discussion accordingly.

9) Please rephrase the following passage: "Raising the possibility that the functions of different Munc13 isoforms in priming LDCVs and SVs are conserved, even though their role in docking is not". This needs to be qualified by the detection issue. There may be two modes of docking as the authors discuss but Munc13s might function in one of these modes similar to that for SVs.

Please also see response to point 3. To confirm that the apparent intrinsic differences in the ability of Munc13-1 and ubMunc13-2 to promote LDCV secretion in chromaffin cells are not merely due to higher expression levels of ubMunc13-2, we have added a supplemental figure (Figure 5—figure supplement 1), where only cells of matched fluorescence intensities were analyzed.

With respect to the two modes of docking, please also see response to point 4. We have modified the corresponding section of the Discussion to state more explicitly that we think that the docking and priming of fusion competent vesicles, i.e. the Munc13-dependent formation of the RRP, is mechanistically similar for SVs and LDCVs.